# FALSA: Fairness-Aware Latent Space Alignment in Vision-Language Models for Medical Image Segmentation

## Abstract

Vision-language models (VLMs) achieve impressive generalization in medical image segmentation but often propagate demographic biases from large-scale pre-training data. These biases manifest not only across single attributes (e.g., gender, race) but also through *intersectional bias*, where overlapping identities amplify disparities, undermining equitable clinical reliability. To address these problems, we introduce **FALSA**, a Fairness-Aware Latent Space Alignment framework that mitigates representational and predictive disparities without sacrificing accuracy. FALSA integrates three components: (1) Demographic-Invariant Contrastive Learning (DICL) to align multimodal embeddings while suppressing group-specific leakage, (2) Adaptive Fairness Calibration (AFC) to adversarially remove demographic cues from latent features, and (3) Unified Cross-Attribute Fairness Loss (UniFairLoss) to jointly reduce intra- and inter-attribute disparities, including intersectional ones. Applied to SAM and SAMed on the Harvard-FairSeg benchmark, FALSA achieves state-of-the-art equity-scaled Dice and IoU scores, reducing disparity indices and performance gaps by up to 70-80% while improving overall segmentation accuracy. Unlike prior models that compromise segmentation accuracy, FALSA advances both fairness and performance, providing a scalable framework for equitable multimodal healthcare AI. Code and Data will be available at https://github.com/.../FALSA.

## 1 Introductions

Recent advancements in vision-language models (VLMs), such as CLIP (Radford et al., 2021), BLIP (Li et al., 2022), Segment Anything Model (SAM) (Kirillov et al., 2023), and customized SAM (SAMed) (Zhang & Liu, 2023), have significantly improved multimodal understanding by learning joint representations from large-scale image-text datasets. These models exhibit exceptional generalization capabilities across diverse tasks, including zero-shot classification, image retrieval, captioning, and image segmentation (Kong et al., 2024; Slyman et al., 2024). Due to their flexibility and minimal task-specific supervision requirements, VLMs are being rapidly adopted in high-impact areas, such as medical imaging, where accurate, scalable, and robust inference is crucial (Wan et al., 2023; Wang et al., 2022; Lin et al., 2023; Bannur et al., 2023).

Despite their strong empirical performance, VLMs are not immune to concerns about fairness (Varma et al., 2024; D'Incà et al., 2024). These models often inherit and amplify the demographic biases present in their pre-training data. On the vision side, under-representation or imbalance in training distributions can lead to systematically degraded performance for certain demographic groups, such as older adults, women, or racial minorities (Howard et al., 2024; Berg et al., 2022; Zhu et al., 2024). On the language side, large-scale textual corpora may contain societal stereotypes

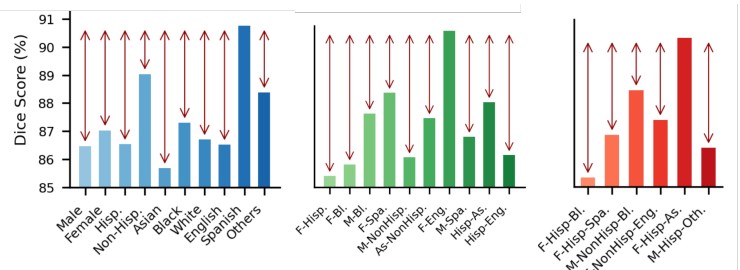

Figure 1: Difference in segmentation performance across sensitive attributes shown by SAMed model on Harvard-FairSeg dataset.

and skewed cultural narratives, which can subtly influence model predictions (Leng et al., 2024). When combined in multimodal settings, these biases can reinforce each other, potentially resulting in harmful or inequitable outcomes (Marani et al., 2024). This compounding effect is particularly hazardous in sensitive domains, such as healthcare, where biased predictions can impact diagnosis, treatment, and trust (Tian et al., 2024; Wan et al., 2023; Luo et al., 2024). As shown in Figure 1, models such as SAMed demonstrate strong performance on certain demographic groups while underperforming on others, exhibiting clear biases in their segmentation outcomes across attributes. The figure also indicates that interactions such as two (middle) or three (right) attributes from different demographic groups contribute significantly to bias in VLMs.

Mitigating bias in VLMs is a complex and multifaceted challenge. Multimodal representations are inherently intertwined and reflect information from diverse and often noisy sources (Gerych et al., 2024; Yin et al., 2023; Seth et al., 2023; Hamidieh et al., 2024). Although previous research has made efforts in fairness-aware learning for unimodal systems, whether vision or language, these methods often fall short when applied to VLMs due to their neglect of the joint structure of cross-modal embeddings (Wang et al., 2025; Tian et al., 2024; Slyman et al., 2024). Moreover, the vast majority of fairness techniques focus on mitigating disparities related to a single sensitive attribute like sex or race, ignoring the intersectional nature of real-world identities (Zhang & Liu, 2023; Zhang et al., 2022). In reality, individuals often fit into multiple demographic categories simultaneously, and fairness measures that focus on just one attribute may overlook the systemic inequalities that are generated by their interactions (Wan et al., 2023). Consequently, achieving fairness across various sensitive attributes simultaneously remains a critical and unresolved research challenge.

In this paper, we present **FALSA**, a Fairness-Aware Latent Space Alignment framework for vision-language models (VLMs), designed to mitigate demographic bias in segmentation tasks without degrading predictive performance. We integrate FALSA into both SAM (standard segmentation) and SAMed (medical segmentation), enhancing them with fairness-aware modules to ensure equitable performance across diverse populations. Unlike prior work focusing on a single sensitive attribute (e.g., gender or race), FALSA jointly addresses *all* sensitive attributes and their intersectional effects. FALSA does not enforce identical latent distributions. Instead, the goal is to remove spurious demographic leakage while preserving clinically relevant variation, achieving consistent performance across groups. The main contributions are summarized as follows:

- **Demographic-Invariant Contrastive Learning (DICL):** We introduce a novel contrastive learning technique that minimizes representation disparities among demographic groups while maintaining semantic alignment.

- **Adaptive Fairness Calibration (AFC):** An adversarial fairness classifier is introduced that dynamically learns to remove demographic-dependent biases from both visual and textual features, ensuring that VLM embeddings remain robust across diverse population groups.

- **Unified Cross-Attribute Fairness Loss (UniFairLoss:)** To ensure fairness across various sensitive attributes at once, we introduce a new approach called UniFairLoss. This method minimizes both intra-attribute disparities and inter-attribute bias correlations by integrating group-level utility deviation penalties with a regularization term based on covariance.

We demonstrate that FALSA effectively reduces bias in VLM embeddings while preserving their downstream task performance, making it a promising approach for ethical AI deployment in real-world multimodal applications.

## 2 RELATED WORK

A concise overview of the related work is provided here due to page limitations; and a more comprehensive review is available in the supplementary material (see Appendix A). Vision-Language Models (VLMs) like CLIP (Radford et al., 2021), BLIP (Li et al., 2022), and SAM (Kirillov et al., 2023) have shown strong generalization across multimodal tasks, but often inherit demographic biases from pretraining data (Zhang et al., 2022; Seth et al., 2023). Several methods aim to mitigate these biases. Adversarial approaches such as ADV (Madras et al., 2018) and adversarial de-biasing (Zhang et al., 2018) obscure demographic information from representations but can introduce training instability or reduce performance. GroupDRO (Sagawa et al., 2019) focuses on subgroup robustness but typically handles only one sensitive attribute. INLP (Ravfogel et al., 2020) and FairCLIP (Luo et al., 2024) improve unimodal fairness but do not extend to segmentation tasks. FairSeg (Tian et al., 2024) is a notable effort in fairness-aware segmentation. It minimizes disparities using equity-scaled metrics but is limited to unimodal inputs and single-attribute fairness. Multimodal models

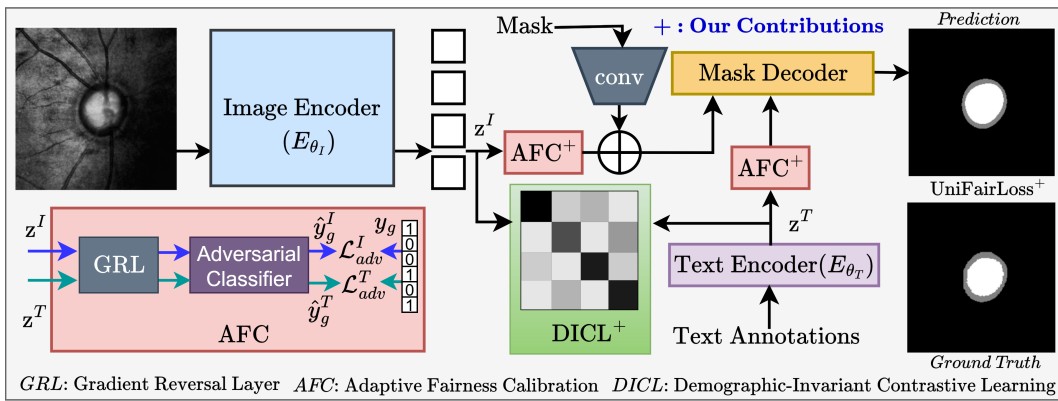

Figure 2: Workflow of the proposed FALSA framework. Here, the modules with a "+" sign indicate our contribution to improving fairness in VLMs.

like MedCLIP (Wang et al., 2022) and PMC-CLIP (Lin et al., 2023) target medical retrieval, without addressing fairness.

FALSA advances this landscape by introducing a unified framework for fairness in vision-language segmentation. It combines demographic-invariant contrastive learning, multi-label adversarial calibration, and a cross-attribute fairness loss. Unlike prior work, FALSA improves both segmentation accuracy and fairness across multiple, intersecting demographic attributes, making it well-suited for equitable multimodal healthcare AI.

## 3 FAIRNESS-AWARE LATENT SPACE ALIGNMENT (FALSA)

FALSA is a fairness-aware framework adapted within SAM and SAMed for experiments, as shown in Figure 2. Here, the modules highlighted with a "+" sign indicate our contribution to this model.

### 3.1 PROBLEM STATEMENT

Let $\mathcal{D} = \{(x_i^I, x_i^T, y_i)\}_{i=1}^N$ denote a multimodal dataset, where $x^I$, $x^T$, and $y$ represent the image, text annotation, and ground truth label, respectively. Each sample is also associated with one or more sensitive demographic attributes. We denote the set of demographic groups as $G = \{g_1, g_2, \ldots, g_m\}$, where each $g \in G$ corresponds to a subgroup such as Male, Female, Asian, Black, and so on.

Vision-language models (VLMs) like SAM align image and text features in a shared embedding space but may encode spurious demographic correlations, leading to biased predictions. Our objective is to enforce demographic invariance while preserving task-relevant information. This can be expressed as the following fairness constraint:

$$\min_f \sum_{g \in G} d(P(y \mid x, g), P(y \mid x)), \tag{1}$$

where $d(\cdot)$ measures the discrepancy between the conditional distribution of predictions for a group and the overall distribution. To achieve this, we build upon transformer-based VLMs, specifically SAM and its medical adaptation SAMed, as the backbone of our approach. Given an image-text pair $(x^I, x^T)$, the encoders map inputs into a shared embedding space:

$$z^I = E_{\theta_I}(x^I), \quad z^T = E_{\theta_T}(x^T), \tag{2}$$

where $E_{\theta_I}$ and $E_{\theta_T}$ are the vision and text encoders, and $z^I, z^T$ are their respective embeddings. We adapt the FALSA framework to align these embeddings in a fairness-aware manner, explicitly mitigating the influence of demographic attributes on model predictions.

### 3.2 DEMOGRAPHIC-INVARIANT CONTRASTIVE LEARNING (DICL)

A major challenge in ensuring fairness in VLMs is the existence of implicit demographic biases within the learned representations, which traditional contrastive learning methods cannot resolve. DICL addresses this problem by explicitly reducing representation disparities across demographic groups while preserving semantic alignment between images and text.

To implement DICL, we define two types of training pairs: **Positive pairs:** Semantically aligned image-text instances $(z_i^I, z_j^T)$ that are deliberately selected from *different demographic groups* ($g_i \neq$

$g_j$). These pairs encourage the model to align representations based on semantic content rather than protected attributes. **Negative pairs:** Instances $(z_i^I, z_j^T)$ that are either semantically dissimilar $(\text{sim}(z_i^I, z_j^T) = 0)$ or drawn from the *same demographic group* $(g_i = g_j)$. These pairs prevent the model from learning spurious associations that reinforce demographic bias. Here, $\text{sim}(z_i^I, z_j^T)$ is a binary function based on cosine similarity, with a threshold (0.5) used to decide whether two samples are semantically aligned:

$$\text{sim}(z_i^I, z_j^T) = \begin{cases} 1, & \text{if } z_i^I \text{ and } z_j^T \text{ are semantically aligned} \\ 0, & \text{otherwise.} \end{cases} \tag{3}$$

The DICL loss is then formulated as:

$$\mathcal{L}_{\text{DICL}} = -\log \frac{\exp\left(\text{sim}(z_i^I, z_j^T)/\tau\right)}{\sum_{k=1}^{N} \mathbb{1}_{[\,g_k = g_i \vee \text{sim}(z_i^I, z_k^T) = 0\,]} \exp\left(\text{sim}(z_i^I, z_k^T)/\tau\right)}, \tag{4}$$

where $\tau$ is a temperature parameter and $\mathbb{1}_{[\cdot]}$ is an indicator function that retains valid negatives (i.e., those that are either semantically dissimilar or from the same group). By excluding hard negatives that are both semantically similar and demographically different, DICL prevents bias amplification and encourages group-invariant alignment.

### 3.3 ADAPTIVE FAIRNESS CALIBRATION (AFC)

We propose adaptive fairness calibration (AFC), a novel adversarial de-biasing framework that promotes demographic invariance in multimodal representation learning. Unlike prior works that focus on a single modality (image or text), AFC explicitly suppresses demographic signals in both visual and textual embeddings. Each demographic group $g \in G$ is represented using a binary membership vector $y_g \in \{0, 1\}^{|G|}$, where $|G|$ is the total number of groups. To remove group-specific information, we introduce an adversarial classifier $C_{\text{adv}}$ for each modality that attempts to predict $y_g$ from the embedding $z \in \{z^I, z^T\}$. The predicted distribution is denoted by $\hat{y}_g = C_{\text{adv}}(z)$. The adversarial loss for each modality is defined as:
$$\mathcal{L}_{\text{adv}} = -\sum_{g \in G} y_g \log \hat{y}_g \tag{5}$$

We incorporate an adversarial de-biasing strategy by placing a gradient reversal layer (GRL) between the encoder and the demographic classifier $C_{\text{adv}}$. The GRL multiplies the gradient of the adversarial loss $\mathcal{L}_{\text{adv}}$ by a negative scalar, penalizing the encoder for retaining demographic information. Classical GRL approaches rely on a fixed adversarial weight $\lambda$, which uniformly enforces demographic invariance across all groups. However, this rigid treatment may over-regularize already invariant groups and under-regularize groups with higher information leakage. To address this, we introduce *group-adaptive gradient reversal*, where the GRL strength is dynamically scaled for each demographic group $g \in G$. Specifically, we define:

$$\lambda_g = \left( \frac{\mathcal{L}_{\text{adv}}^{(g)}}{\max_{h \in G} \mathcal{L}_{\text{adv}}^{(h)}} \right)^{\gamma}, \tag{6}$$

where $\mathcal{L}_{\text{adv}}^{(g)}$ denotes the group-wise adversarial loss, estimated using an exponential moving average (EMA) for stability, and $\gamma \geq 1$ is a temperature parameter that controls the sharpness of scaling. This design adaptively increases adversarial pressure on groups exhibiting stronger demographic leakage, while preventing excessive regularization of groups that are already obfuscated. The final fairness-aware adversarial loss aggregates group-specific terms for both vision and text modalities:

$$\mathcal{L}_{\text{AFC}} = \sum_{g \in G} \left( \lambda_g^I \cdot \mathcal{L}_{\text{adv}}^{I,(g)} + \lambda_g^T \cdot \mathcal{L}_{\text{adv}}^{T,(g)} \right), \tag{7}$$

Because AFC operates on a multi-label demographic vector, its adversarial head jointly suppresses leakage from race, gender, ethnicity, and language, allowing the model to mitigate intersectional bias by targeting cues that emerge only from the co-occurrence of multiple attributes. During backpropagation, the GRL inverts these gradients to discourage the encoder from encoding group-specific signals in either modality. By adaptively calibrating adversarial pressure across all subgroups, AFC enforces demographic invariance at both intra-group and inter-group levels.

### 3.4 UNIFIED CROSS-GROUP FAIRNESS LOSS (UNIFAIRLOSS)

Fairness in vision-language models is often evaluated separately across demographic groups, such as Male, Female, Asian, Hispanic. However, treating groups in isolation overlooks intersectional

disparities (e.g., Hispanic females or non-English-speaking minorities). We propose a UniFairLoss that jointly enforces consistency within groups and across group intersections. Let $U_g$ denote the task performance metric (e.g., Dice, IoU (intersection over union)) for group $g \in G$, with overall mean $\bar{U} = \frac{1}{|G|} \sum_{g \in G} U_g$. The normalized deviation is $\delta_g = \frac{U_g - \bar{U}}{\bar{U}}$.

**Intra- and cross-group penalties.** We introduce two fairness terms: the intra-group penalty

$$\mathcal{L}_{\text{fair}} = \sum_{g \in G} \delta_g^2, \tag{8}$$

which measures how far each group's performance deviates from the overall mean, and the cross-group penalty

$$\mathcal{L}_{\text{cross}} = \frac{1}{|G|^2} \sum_{i=1}^{|G|} \sum_{j=1}^{|G|} \text{Cov}(\delta_i, \delta_j), \tag{9}$$

where $\text{Cov}(\delta_i, \delta_j)$ is the covariance between group deviations, meaning it captures whether the performance gaps of groups $i$ and $j$ tend to increase or decrease together. The first term reduces variance across groups, while the second prevents systematic co-failures across intersections. In practice, group utilities $U_g$ and covariances $\text{Cov}(\delta_i, \delta_j)$ are estimated using EMA-smoothed, cross-batch aggregated statistics, with group-balanced sampling and gradient clipping applied during optimization. These stabilization steps ensure low-variance and reliable fairness signals even for minority or sparsely represented intersectional subgroups.

**Unified objective.** The final UniFairLoss integrates fairness penalties with the base task loss:

$$\mathcal{L}_{\text{UniFair}} = \underbrace{\mathcal{L}_{\text{task}}}_{\text{segmentation}} + \underbrace{\alpha \mathcal{L}_{\text{fair}}}_{\text{intra-group fairness}} + \underbrace{\beta \mathcal{L}_{\text{cross}}}_{\text{cross-group fairness}}, \tag{10}$$

where $\mathcal{L}_{\text{task}}$ is the task-specific objective (CrossEntropy + Dice), and $\alpha, \beta$ balance fairness against accuracy. By coupling intra-group consistency with cross-group correlation penalties, UniFairLoss mitigates both individual and intersectional disparities, ensuring equitable performance across all demographic subgroups while preserving task fidelity. The final loss function to supervise the model end-to-end is the summation of all losses computed in the earlier subsections as $\mathcal{L} = \mathcal{L}_{\text{DICL}} + \mathcal{L}_{\text{adv}} + \mathcal{L}_{\text{UniFair}}$. The combination of all losses allows the model to eliminate biases for all attributes without compromising segmentation performance.

**FALSA's Position.** FALSA fundamentally differs from prior fairness methods by overcoming their key limitations in multimodal VLMs. Earlier approaches rely on static demographic pairings, unimodal contrastive losses, and uniform negative sampling. In contrast, DICL introduces semantic cross-demographic positive selection and demographic-conditioned negative filtering, ensuring alignment is driven by anatomical similarity rather than demographic co-occurrence, avoiding fairness and performance conflicts and preventing representational collapse. AFC addresses the instability of fixed-weight adversarial debiasing by using leakage-aware, EMA-smoothed adaptive scaling across race, gender, ethnicity, and language, enabling robust intersectional mitigation. Finally, UniFairLoss extends beyond variance reduction with a covariance-based co-failure penalty that suppresses correlated error patterns across demographic groups. By computing subgroup statistics via EMA and cross-batch aggregation, UniFairLoss provides stable, low-variance fairness gradients even for small subgroups.

## 4 EXPERIMENTS

**Dataset:** We use Harvard-FairSeg, a large-scale benchmark for fairness evaluation in medical image segmentation. It includes 10,000 scanning laser ophthalmoscopy (SLO) fundus images from unique patients, each with expert-refined optic cup and rim masks derived from 3D optical coherence tomography (OCT) scans. Each sample contains two textual medical notes, one written by a clinical curator and one generated by ChatGPT, supporting multimodal vision-language studies. Demographic annotations include gender, race, ethnicity, language, and marital status. In our experiments, we use the first four categories and encode them as a 10-dimensional binary group-membership (multi-hot) vector. Each dimension corresponds to a specific group (e.g., Female, Asian, Non-Hispanic, Spanish), and multiple entries may be active for a single individual. For example, a female, Asian, non-Hispanic, Spanish speaking individual is represented as $[1, 0, 1, 0, 0, 1, 0, 0, 1, 0]$. The dataset features strong racial diversity, including 919 Asian, 1,473 black and 7,608 white patients, which enables robust intersectional fairness analysis for the detection of glaucoma. The detailed dataset description is provided in the supplementary material (see Appendix B). We also conduct a detailed experiment on a secondary dataset, shown in Appendix D.1.

**Evaluation Metrics:** We evaluate segmentation using Dice to measure overlap, and the intersection-over-union (IoU) between predicted and ground truth masks, respectively. To assess fairness, we report Equity-Scaled (ES) Dice and IoU, defined as ES-Metric $= \frac{\bar{M}}{1+\sum_{g \in G}|\bar{M}-M_g|}$, where $M_g$ is the score for group $g$ and $\bar{M}$ is the overall mean. Lower disparity yields values closer to the standard Dice/IoU. We also report group standard deviation (GSD), disparity index (DI), and relative performance gap (RPG), which capture inter-group performance variance, average absolute deviation, and normalized subgroup gap, respectively. The supplementary materials section contains the detailed evaluation metrics (see Appendix C).

**Training Parameters:** We employed consistent training configurations across all compared segmentation models. The model was trained for a maximum of 300 epochs, with early stopping based on the validation Dice score. The early stopping patience was set to 30 epochs, and the minimum improvement was set to 0.0005. For optimization, we used the AdamW optimizer with exponential learning rate decay. The learning rate was set to 0.001, with a weight decay of $1 \times 10^{-4}$ and a momentum of 0.9. All models were trained using a batch size of 16. We conducted each experiment five times and reported the mean in this study.

**Comparative Models:** We compare FALSA's performance with Adversarially Fair Representations (ADV) (Madras et al., 2018), which reduce biases by making sensitive attributes difficult to predict, and the Group Distributionally Robust Optimization (GroupDRO) (Sagawa et al., 2019) model, which minimizes maximum training loss to improve fairness. Additionally, we benchmark against FEBS (Tian et al., 2024), which reweights the loss function using upper error bounds for each identity group through the Segment Anything Model (SAM).

## 5 RESULTS

We evaluate FALSA and competing state-of-the-art (SOTA) methods on the Harvard-FairSeg dataset for Cup and Rim segmentation in retinal images using curator-provided medical notes.

Table 1: Performance of all compared fairness models regarding Dice (↑) scores (%), Equity-Scaled (ES) Dice, and Group Standard Deviation (GSD) (↓) for cup and rim regions of the Harvard-FairSeg dataset. The best values are highlighted in **bold** within each base model per region.

| Region | Method | ES-Dice | Dice | Female | Male | Hisp. | Non-Hisp. | Asian | Black | White | English | Spanish | Others | GSD |
|---|---|---|---|---|---|---|---|---|---|---|---|---|---|---|
| Cup | SAMed | 84.50 | 86.71 | 87.03 | 86.47 | 86.53 | **89.04** | 85.68 | 87.30 | 86.70 | 86.52 | 90.77 | 88.38 | 1.63 |
| | SAMed+ADV | 84.83 | 86.82 | 86.75 | 86.61 | 86.61 | 88.83 | 85.90 | 87.05 | 87.08 | 86.68 | **91.31** | 88.20 | 1.63 |
| | SAMed+GroupDRO | 85.13 | 86.92 | 86.70 | 86.72 | 86.82 | 88.55 | 85.83 | 87.04 | 87.06 | 86.84 | 90.85 | 88.49 | 1.47 |
| | SAMed+FEBS | 85.12 | 86.86 | **87.56** | 87.18 | 87.04 | 88.24 | 85.87 | **87.08** | 86.72 | 86.70 | 90.34 | **87.94** | 1.29 |
| | **SAMed+FALSA** | **86.82** | **87.01** | 87.49 | **87.21** | **87.50** | 88.30 | **87.31** | 87.26 | **87.20** | **87.17** | 89.97 | 88.76 | **0.88** |
| | SAM | 83.59 | 85.81 | 86.13 | 85.58 | 85.63 | **88.04** | 83.70 | 85.89 | 86.03 | 85.69 | 90.72 | 86.31 | 1.76 |
| | SAM+ADV | 82.45 | 84.47 | 84.48 | 84.44 | 84.04 | 86.61 | 83.46 | 85.17 | 85.26 | 83.96 | 89.33 | 84.38 | 1.63 |
| | SAM+GroupDRO | 83.19 | 85.55 | 86.28 | 85.41 | 85.68 | 87.48 | 82.97 | 85.69 | 85.64 | 84.98 | 89.44 | 86.71 | 1.60 |
| | SAM+FEBS | 83.69 | 85.79 | 86.14 | 85.94 | 86.01 | 87.61 | 83.48 | 85.84 | 85.84 | 85.67 | 90.34 | 86.62 | 1.65 |
| | **SAM+FALSA** | **85.72** | **86.13** | **86.82** | **86.65** | **86.47** | 86.51 | **85.53** | **86.76** | **86.89** | **86.68** | 88.08 | **86.81** | **0.59** |
| Rim | SAMed | 79.40 | 82.91 | 82.52 | 83.19 | 82.77 | 83.97 | 78.90 | 77.58 | 84.44 | 83.05 | 85.34 | 79.89 | 2.44 |
| | SAMed+ADV | 79.38 | 82.91 | 82.63 | 83.42 | 83.08 | 84.16 | 78.01 | 76.91 | 83.95 | 83.07 | 85.28 | 80.15 | 2.64 |
| | SAMed+GroupDRO | 79.84 | 83.08 | 82.74 | **83.53** | 82.84 | 83.90 | 79.52 | 77.48 | 84.54 | 83.22 | 84.93 | 80.65 | 2.29 |
| | SAMed+FEBS | 79.88 | 83.21 | 82.89 | 83.38 | 83.49 | **84.08** | 79.52 | 77.89 | **84.73** | 83.28 | 85.11 | 79.92 | 2.34 |
| | **SAMed+FALSA** | **83.48** | **84.01** | **83.35** | 83.46 | **83.58** | 83.65 | **82.59** | **82.43** | 83.90 | **83.31** | 85.35 | **83.13** | **0.80** |
| | SAM | 76.62 | 80.27 | 79.94 | 80.51 | 80.14 | 81.57 | 75.57 | 74.07 | 82.06 | 80.40 | **82.65** | 77.33 | 2.70 |
| | SAM+ADV | 76.12 | 79.71 | 78.79 | 80.05 | 79.29 | 80.15 | 75.13 | 73.86 | 81.87 | 80.03 | 80.64 | 76.01 | 2.51 |
| | SAM+GroupDRO | 76.82 | 80.03 | 80.00 | 80.30 | 80.36 | 80.01 | **75.70** | 73.29 | 81.80 | 79.67 | 81.57 | 77.28 | 2.59 |
| | SAM+FEBS | 76.78 | 80.21 | 80.32 | 80.24 | 80.43 | 81.40 | 75.79 | 74.25 | **82.30** | 80.09 | 82.06 | 77.61 | 2.56 |
| | **SAM+FALSA** | **80.24** | **81.03** | **80.58** | **80.61** | **81.46** | **81.59** | 81.93 | 81.58 | 82.04 | 80.82 | 81.11 | **80.67** | **0.52** |

**Dice Performance.** Table 1 shows that FALSA improves both accuracy and fairness across all settings. For *Cup* segmentation, SAMed+FALSA achieves the highest Dice (87.01%) and ES-Dice (86.82%) with the lowest GSD (0.88), while SAM+FALSA also leads with Dice/ES-Dice of 86.13%/85.72% and GSD of 0.59. For *Rim*, SAMed+FALSA attains Dice/ES-Dice of 84.01%/83.48% with GSD of 0.80; SAM+FALSA achieves 81.03%/80.24% Dice/ES-Dice with GSD of 0.52. These results are consistently higher and more equitable than all baselines.

**IoU Performance.** Table 2 confirms similar trends. SAMed+FALSA yields IoU/ES-IoU of 79.34%/78.86% (Cup) and 72.96%/72.08% (Rim), outperforming others by 1-3%, with lowest GSD (0.73, 0.36). SAM+FALSA achieves 77.18%/76.74% (Cup) and 71.83%/70.08% (Rim), with GSD of 0.52 and 0.87, respectively.

Table 2: Performance of all compared fairness models regarding IoU (↑) scores (%) and GSD (↓) for cup and rim regions. The best values are highlighted in **bold** within each base model per region.

| Region | Method | ES-IoU | IoU | Male | Female | Hisp. | Non-Hisp. | Asian | Black | White | English | Spanish | Others | GSD |
|---|---|---|---|---|---|---|---|---|---|---|---|---|---|---|
| Cup | SAMed | 75.64 | 78.13 | 77.83 | 78.55 | 77.90 | **81.00** | 76.88 | 79.05 | 78.08 | 77.91 | 83.38 | **80.01** | 2.01 |
| | SAMed+ADV | 75.86 | 78.22 | 77.91 | 78.20 | 77.91 | 80.80 | 77.09 | 78.82 | 78.46 | 78.08 | **84.32** | 79.82 | 2.13 |
| | SAMed+GroupDRO | 76.24 | 78.33 | 78.04 | 78.14 | 78.19 | 80.44 | 77.11 | 78.86 | 78.42 | 78.25 | 83.60 | 80.19 | 1.89 |
| | SAMed+FEBS | 76.16 | 78.26 | 78.51 | 78.79 | 79.04 | 80.70 | 77.08 | 78.82 | 78.04 | 78.10 | 82.68 | 79.37 | 1.66 |
| | **SAMed+FALSA** | **78.86** | **79.34** | **79.93** | **80.07** | **81.12** | 80.92 | **79.24** | **79.18** | **79.15** | **79.34** | 80.26 | 79.41 | **0.73** |
| | SAM | 73.81 | 76.32 | 76.08 | 76.64 | 76.08 | **79.26** | 73.77 | 76.76 | 76.51 | 76.16 | **82.66** | 76.92 | 2.24 |
| | SAM+ADV | 72.28 | 74.57 | 74.61 | 74.50 | 73.94 | 77.22 | 73.60 | 75.82 | 75.40 | 74.01 | 80.90 | 74.76 | 2.07 |
| | SAM+GroupDRO | 73.49 | 75.92 | 75.83 | 76.75 | 76.07 | 78.35 | 73.32 | 76.29 | 75.95 | 75.21 | 80.93 | 77.05 | 1.90 |
| | SAM+FEBS | 73.87 | 76.23 | 76.05 | 76.56 | 76.15 | 78.64 | 73.47 | 76.50 | 76.13 | 76.04 | 82.09 | 77.19 | 2.11 |
| | **SAM+FALSA** | **76.74** | **77.18** | **77.11** | **77.24** | **77.98** | 78.32 | **77.28** | **77.43** | **77.52** | **77.79** | 78.86 | **77.84** | **0.52** |
| Rim | SAMed | 68.74 | 72.17 | 72.52 | 71.69 | 72.03 | 73.07 | 67.43 | 65.87 | 73.99 | 72.34 | **74.68** | 68.71 | 2.75 |
| | SAMed+ADV | 68.69 | 72.12 | 72.76 | 71.81 | 72.41 | **73.42** | 66.35 | 64.98 | 73.25 | 72.31 | 74.63 | 69.00 | 3.02 |
| | SAMed+GroupDRO | 69.21 | 72.37 | **72.92** | 71.98 | 72.08 | 72.98 | 68.22 | 65.68 | 74.10 | 72.53 | 74.11 | 69.54 | 2.58 |
| | SAMed+FEBS | 69.15 | 72.50 | 72.74 | 72.23 | 72.78 | 73.29 | 68.25 | 66.20 | **74.39** | 72.63 | 74.36 | 68.65 | 2.66 |
| | **SAMed+FALSA** | **72.08** | **72.96** | 72.91 | **72.61** | **73.28** | 73.32 | **72.64** | **72.57** | 73.07 | **72.91** | 72.43 | **72.26** | **0.36** |
| | SAM | 66.70 | 70.06 | 70.36 | 69.65 | 69.95 | **71.15** | 64.60 | 62.91 | 72.13 | 70.21 | **72.50** | 66.98 | 3.03 |
| | SAM+ADV | 66.17 | 69.45 | 69.99 | 68.34 | 68.90 | 69.58 | 64.16 | 62.82 | 71.88 | **69.87** | 70.17 | 65.65 | 2.78 |
| | SAM+GroupDRO | 66.71 | 69.80 | 70.16 | 69.77 | 70.28 | 69.46 | 64.83 | 61.99 | 71.87 | 69.25 | 71.00 | 66.55 | 2.94 |
| | SAM+FEBS | 66.99 | 70.11 | 69.78 | 69.53 | 69.97 | 70.89 | 64.85 | 63.20 | **72.35** | 69.80 | 71.65 | 67.24 | 2.80 |
| | **SAM+FALSA** | **70.08** | **71.83** | **70.39** | **70.87** | **71.13** | 70.97 | **69.17** | **69.79** | 70.98 | 68.97 | 69.75 | **68.68** | **0.87** |

**Fairness Indicators.** Table 3 reports DI and RPG reductions across groups. For Cup, SAMed+FALSA reduces DI from 1.17 to 0.72 (Dice) and 1.44 to 0.60 (IoU); SAM+FALSA lowers DI from 1.20 to 0.59 (Dice) and 1.55 to 0.52 (IoU), yielding 40-50% fairness gains. Rim improvements are even larger: DI drops from 2.03 to 0.76 (SAMed) and 2.26 to 0.48 (SAM) for Dice, yielding over 70-80% reduction. RPG follows the same pattern, confirming equitable outcomes. FALSA consistently achieves the best segmentation and fairness performance, evidenced by top Dice/IoU, ES metrics, lowest GSD, and minimal DI/RPG.

Table 3: Comparative performance of fairness models regarding disparity index (DI) and relative performance gap (RPG) for Dice and IoU across Cup and Rim.

| Region | Method | Dice | | IoU | |
|---|---|---|---|---|---|
| | | DI↓ | RPG↓ | DI↓ | RPG↓ |
| Cup | SAMed | 1.17 | 5.61 | 1.44 | 7.80 |
| | SAMed+ADV | 1.17 | 5.93 | 1.50 | 8.57 |
| | SAMed+GroupDRO | 1.08 | 5.53 | 1.37 | 7.76 |
| | SAMed+FEBS | 0.84 | 4.95 | 1.08 | 6.77 |
| | **SAMed+FALSA** | **0.72** | **3.11** | **0.60** | **2.43** |
| | SAM | 1.20 | 7.82 | 1.55 | 10.89 |
| | SAM+ADV | 1.18 | 6.65 | 1.50 | 9.14 |
| | SAM+GroupDRO | 1.16 | 7.32 | 1.36 | 9.52 |
| | SAM+FEBS | 1.10 | 7.68 | 1.45 | 10.63 |
| | **SAM+FALSA** | **0.59** | **2.94** | **0.52** | **2.25** |
| Rim | SAMed | 2.03 | 9.09 | 2.34 | 11.80 |
| | SAMed+ADV | 2.23 | 9.81 | 2.59 | 12.93 |
| | SAMed+GroupDRO | 1.87 | 8.77 | 2.16 | 11.38 |
| | SAMed+FEBS | 1.99 | 8.48 | 2.31 | 11.01 |
| | **SAMed+FALSA** | **0.76** | **3.42** | **0.34** | **1.45** |
| | SAM | 2.26 | 10.51 | 2.53 | 13.80 |
| | SAM+ADV | 2.15 | 9.91 | 2.36 | 13.15 |
| | SAM+GroupDRO | 2.14 | 10.53 | 2.44 | 14.35 |
| | SAM+FEBS | 2.14 | 9.90 | 2.30 | 13.19 |
| | **SAM+FALSA** | **0.48** | **1.80** | **0.80** | **3.60** |

**Impact of Each Fairness Component:** We evaluate the incremental contributions of AFC, DICL, and UniFairLoss via ablation studies on SAMed and SAM for both Cup and Rim regions, using Dice, IoU, and their equity-scaled (ES) versions (Table 4).

**Cup Region.** Adding DICL to baseline SAMed raises ES-Dice/ES-IoU from 84.50%/75.64% to 85.37%/76.45%; in SAM, from 82.59%/72.81% to 83.87%/74.19%, yielding 0.8-1.5% gains from demographic-invariant alignment. Adding AFC provides further boosts: SAMed reaches 86.25%/77.13% and SAM 84.65%/75.38% (additional 0.7-1.0% over DICL). With all three modules enabled, SAMed

Table 4: Quantify the impact of each fairness component of FALSA across methods regarding Dice (↑), IoU (↑), and their ES variants.

| Region | Model | DICL | AFC | UniFairLoss | ES-Dice | Overall Dice | ES-IoU | Overall IoU |
|---|---|---|---|---|---|---|---|---|
| Cup | SAMed | | | | 84.50 | 86.71 | 75.64 | 78.13 |
| | SAMed | ✓ | | | 85.37 | 86.89 | 76.45 | 78.78 |
| | SAMed | ✓ | ✓ | | 86.25 | 86.92 | 77.13 | 79.29 |
| | **SAMed** | ✓ | ✓ | ✓ | **86.82** | **87.01** | **78.86** | **79.34** |
| | SAM | | | | 82.59 | 84.81 | 72.81 | 75.32 |
| | SAM | ✓ | | | 83.87 | 84.93 | 74.19 | 75.65 |
| | SAM | ✓ | ✓ | | 84.65 | 85.04 | 75.38 | 75.88 |
| | **SAM** | ✓ | ✓ | ✓ | **85.72** | **86.13** | **76.74** | **77.18** |
| Rim | SAMed | | | | 79.40 | 82.91 | 68.74 | 72.17 |
| | SAMed | ✓ | | | 80.57 | 83.43 | 69.74 | 72.39 |
| | SAMed | ✓ | ✓ | | 81.69 | 84.68 | 70.59 | 72.62 |
| | **SAMed** | ✓ | ✓ | ✓ | **83.48** | **84.01** | **72.08** | **72.96** |
| | SAM | | | | 75.62 | 79.27 | 63.70 | 67.06 |
| | SAM | ✓ | | | 76.88 | 79.33 | 64.19 | 67.83 |
| | SAM | ✓ | ✓ | | 78.61 | 79.69 | 65.43 | 67.97 |
| | **SAM** | ✓ | ✓ | ✓ | **80.24** | **81.03** | **70.08** | **71.83** |

peaks at 86.82%/78.86% (total +2.32/+3.22% over baseline) and SAM at 85.72%/76.74% (2-3% increase), confirming the additive benefit of the full FALSA configuration.

**Rim Region.** A similar progression is observed: SAMed baseline ES-Dice/ES-IoU 79.40%/68.74% $\rightarrow$ 80.57%/70.09% (DICL) $\rightarrow$ 81.69%/71.48% (AFC) $\rightarrow$ 83.48%/72.08% (full FALSA). SAM follows 75.62%/63.70% $\rightarrow$ 77.80%/66.24% $\rightarrow$ 79.18%/68.92% $\rightarrow$ 80.24%/70.08%, yielding 3-4% overall improvements. These consistent gains across subgroups highlight the importance of all fairness modules.

**Feature De-biasing.** Figure 3 visualizes the image/text embeddings before and after FALSA. Pre-de-biasing, features cluster strongly by demographic attributes, indicating representational bias; post-FALSA, clusters become more dispersed and mixed, demonstrating effective removal of attribute-specific signals and fairer, attribute-invariant embeddings.

**Impact of FALSA on Intersectional Demographic Bias:** Assessing VLM fairness through intersectional demographics is essential, as actual patients encompass multifaceted and overlapping identities such as race, gender, ethnicity, and language. To thoroughly evaluate fairness, we developed intersectional subgroups by selecting attributes from various demographic categories. This approach reflects real-world complexities and investigates whether FALSA can mitigate compounded biases among diverse and intersecting populations.

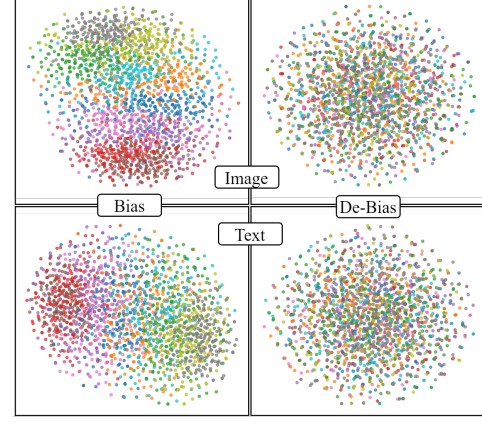

Figure 3: t-Distributed Stochastic Neighbor Embedding (t-SNE) plots of encoded image (top) and text (bottom) features, showing biased (left) vs. de-biased (right) representations after FALSA.

Table 5: Performance (Dice and IoU) of Baseline and FALSA-enhanced models for 2-attribute intersection from 2 different groups. The most biased combinations are reported due to space limitations.

| Region | Method | F-Hisp. | M-NonHisp. | F-Bl. | M-Bl. | F-Eng. | F-Spa. | M-Spa. | Hisp-As. | Hisp-Eng. | Bl-Eng. | Wh-Spa. | GSD |
|---|---|---|---|---|---|---|---|---|---|---|---|---|---|
| | | | | | | Dice | | | | | | | |
| Cup | SAMed | 85.06 | 86.02 | 87.14 | 84.38 | 89.34 | 84.11 | 83.69 | 88.37 | 89.67 | 88.12 | 85.33 | 2.15 |
| | SAMed+FALSA | 88.68 | 88.74 | 88.59 | 89.14 | 89.02 | 88.65 | 88.79 | 88.93 | 89.12 | 87.96 | 87.94 | 0.41 |
| | SAM | 84.36 | 82.39 | 83.61 | 84.31 | 81.67 | 80.98 | 83.47 | 85.14 | 87.31 | 86.17 | 86.09 | 1.97 |
| | SAM+FALSA | 85.81 | 85.76 | 86.17 | 85.69 | 85.97 | 86.06 | 86.10 | 85.88 | 86.13 | 84.98 | 85.19 | 0.39 |
| Rim | SAMed | 77.81 | 78.95 | 79.21 | 77.44 | 78.62 | 79.87 | 78.14 | 77.96 | 79.02 | 77.68 | 78.43 | 0.75 |
| | SAMed+FALSA | 79.48 | 80.52 | 79.61 | 79.24 | 80.05 | 79.33 | 79.72 | 80.18 | 79.54 | 79.67 | 79.41 | 0.39 |
| | SAM | 61.24 | 59.83 | 63.17 | 62.42 | 65.21 | 60.98 | 64.53 | 67.14 | 66.32 | 68.15 | 65.89 | 2.74 |
| | SAM+FALSA | 66.74 | 67.12 | 66.53 | 67.45 | 66.89 | 67.24 | 66.78 | 67.41 | 67.16 | 66.84 | 66.92 | 0.29 |
| | | | | | | IoU | | | | | | | |
| Cup | SAMed | 76.81 | 78.23 | 77.45 | 79.12 | 81.34 | 78.76 | 80.12 | 77.89 | 81.02 | 79.67 | 80.43 | 1.49 |
| | SAMed+FALSA | 79.68 | 80.12 | 79.79 | 80.56 | 79.92 | 80.33 | 80.14 | 80.01 | 80.45 | 79.87 | 79.94 | 0.29 |
| | SAM | 73.48 | 74.23 | 75.89 | 76.51 | 77.36 | 75.14 | 78.29 | 79.67 | 78.11 | 80.02 | 77.94 | 2.12 |
| | SAM+FALSA | 76.45 | 77.32 | 76.78 | 77.11 | 77.89 | 76.92 | 77.45 | 76.83 | 77.12 | 76.97 | 77.01 | 0.38 |
| Rim | SAMed | 67.85 | 69.23 | 68.74 | 70.12 | 71.48 | 70.67 | 72.03 | 69.89 | 73.12 | 71.45 | 72.34 | 1.63 |
| | SAMed+FALSA | 72.56 | 73.41 | 72.89 | 73.15 | 73.02 | 72.74 | 73.18 | 72.96 | 73.22 | 72.83 | 72.91 | 0.24 |
| | SAM | 60.34 | 62.41 | 61.78 | 63.25 | 65.49 | 64.03 | 66.12 | 67.48 | 68.15 | 69.02 | 67.84 | 2.91 |
| | SAM+FALSA | 66.45 | 66.92 | 66.74 | 67.12 | 66.88 | 66.67 | 67.03 | 66.91 | 67.18 | 66.85 | 66.94 | 0.21 |

We construct cross-category groups, such as two-attribute and three-attribute intersectional groups, and the impact of FALSA is shown in Tables 5 and 6, respectively. Baseline models, SAMed and SAM, exhibited considerable GSD in two-attribute combinations. SAMed showed GSDs of 2.15 (Cup Dice) and 1.49 (Cup IoU), while SAM exhibited values of 1.97 and 2.12, respectively, indicating significant performance variability across subgroups. FALSA reduced these GSDs to 0.41 and 0.29 (SAMed) and 0.39 and 0.38 (SAM), reflecting an over 80% reduction (Table 5). Similarly, in three-attribute settings (Table 6), SAMed consistently exhibits GSD reductions from 2.13 to 0.39 (Cup Dice) and from 1.61 to 0.30 (Rim IoU). SAM's GSD in Dice (Cup) decreased from 1.90 to 0.37, and in IoU (Rim) from 2.82 to 0.24 after applying FALSA. Importantly, FALSA achieved these fairness improvements without compromising the mean Dice and IoU scores across groups.

**Component-wise ablation of UNIFAIRLOSS:** To assess the individual and combined contributions of UniFairLoss components, we perform an ablation study across task-only, intra-attribute, cross-attribute, and full configurations of this loss function. As shown in Table 7, incorporating both intra- and cross-attribute fairness terms leads to consistent improvements in equity-scaled metrics. For example, in the Cup region with SAM+FALSA, ES-

Table 6: Performance (Dice and IoU) of Baseline and FALSA-enhanced models for 3-attribute intersection from 3 different groups.The most biased combinations are reported due to space limitations.

| Region | Method | F-Hisp-Bl. | F-Hisp-Spa. | M-NonHisp-Bl. | F-NonHisp-Eng. | F-Hisp-As. | M-Hisp-Oth. | GSD |
|---|---|---|---|---|---|---|---|---|
| | | | | Dice | | | | |
| Cup | SAMed | 83.12 | 85.43 | 84.67 | 86.78 | 89.12 | 87.45 | 2.13 |
| | SAMed+FALSA | 88.56 | 88.92 | 89.14 | 89.67 | 89.45 | 89.23 | 0.39 |
| | SAM | 81.34 | 82.45 | 83.67 | 84.56 | 86.23 | 85.78 | 1.90 |
| | SAM+FALSA | 66.45 | 67.12 | 66.74 | 67.45 | 67.28 | 66.91 | 0.37 |
| Rim | SAMed | 76.34 | 77.45 | 78.12 | 78.67 | 79.89 | 79.23 | 1.27 |
| | SAMed+FALSA | 79.78 | 80.12 | 80.45 | 80.67 | 80.89 | 79.92 | 0.44 |
| | SAM | 63.45 | 64.78 | 65.12 | 66.34 | 68.12 | 67.45 | 1.75 |
| | SAM+FALSA | 66.58 | 67.12 | 66.89 | 67.34 | 67.01 | 66.78 | 0.27 |
| | | | | IoU | | | | |
| Cup | SAMed | 76.81 | 78.23 | 77.45 | 79.12 | 81.34 | 80.12 | 1.70 |
| | SAMed+FALSA | 79.68 | 80.12 | 79.79 | 80.56 | 80.14 | 79.94 | 0.31 |
| | SAM | 73.48 | 74.23 | 75.89 | 76.51 | 78.29 | 77.94 | 1.94 |
| | SAM+FALSA | 76.45 | 77.32 | 76.78 | 77.11 | 77.45 | 77.01 | 0.36 |
| Rim | SAMed | 67.85 | 69.23 | 68.74 | 70.12 | 72.03 | 71.45 | 1.61 |
| | SAMed+FALSA | 72.56 | 73.41 | 72.89 | 73.15 | 73.18 | 72.91 | 0.30 |
| | SAM | 60.34 | 62.41 | 61.78 | 63.25 | 66.12 | 67.84 | 2.82 |
| | SAM+FALSA | 66.45 | 66.92 | 66.74 | 67.12 | 67.03 | 66.94 | 0.24 |

Dice rises from 83.12 (task only) to 85.72 with full UniFairLoss, and ES-IoU increases from 74.01 to 76.74. Similar gains are observed with SAMed+FALSA, reaching 86.82 ES-Dice and 78.86 ES-IoU. In the Rim region, SAMed+FALSA improves from 80.86 to 83.48 in ES-Dice and from 69.42 to 72.08 in ES-IoU. These results validate that jointly enforcing intra- and cross-attribute fairness enhances segmentation robustness and demographic equity. Additional ablation studies were conducted to further demonstrate the effectiveness of FALSA in ensuring fairness in segmentation performance, with detailed results provided in Appendix D.

**Clinical implications.** Beyond quantitative gains, FALSA's fairness benefits are vital for equitable healthcare. In medical image segmentation, demographic disparities can lead to unequal diagnostics and treatment. FALSA reduces subgroup errors, lowering the underdiagnosis risk in groups like those with darker skin or minority backgrounds. This promotes consistent disease and lesion detection, essential for surgical planning, radiotherapy, and monitoring. FALSA's fairness improvements boost both technical and clinical decision-making reliability.

Table 7: Ablation Study of UniFairLoss (ES-Dice, Dice, ES-IoU, IoU) across Configurations

| Region | Method | UniFair Config | ES-Dice | Dice | ES-IoU | IoU |
|---|---|---|---|---|---|---|
| Cup | SAM+FALSA | Task Only | 83.12 | 83.56 | 74.01 | 74.65 |
| | | Task + Intra | 84.45 | 84.88 | 75.30 | 75.94 |
| | | Task + Cross | 84.92 | 85.30 | 75.70 | 76.20 |
| | | **UniFairLoss** | **85.72** | **86.13** | **76.74** | **77.18** |
| | SAMed+FALSA | Task Only | 84.30 | 84.72 | 75.42 | 76.04 |
| | | Task + Intra | 85.65 | 86.02 | 76.71 | 77.20 |
| | | Task + Cross | 85.91 | 86.36 | 77.06 | 77.58 |
| | | **UniFairLoss** | **86.82** | **87.01** | **78.86** | **79.34** |
| Rim | SAM+FALSA | Task Only | 77.28 | 77.93 | 67.34 | 68.76 |
| | | Task + Intra | 78.64 | 79.25 | 68.73 | 70.10 |
| | | Task + Cross | 78.97 | 79.61 | 69.05 | 70.40 |
| | | **UniFairLoss** | **80.24** | **81.03** | **70.08** | **71.83** |
| | SAMed+FALSA | Task Only | 80.86 | 81.33 | 69.42 | 70.30 |
| | | Task + Intra | 82.04 | 82.52 | 70.67 | 71.52 |
| | | Task + Cross | 82.42 | 82.94 | 71.06 | 71.88 |
| | | **UniFairLoss** | **83.48** | **84.01** | **72.08** | **72.96** |

## 6 CONCLUSION

In this work, we introduced FALSA, a unified fairness-aware latent space alignment framework that reduces demographic bias in vision-language segmentation without compromising predictive performance. By simultaneously implementing Demographic-Invariant Contrastive Learning, Adaptive Fairness Calibration, and a Unified Cross-Attribute Fairness Loss, FALSA consistently enhances both accuracy and fairness across single and intersectional demographic attributes. Extensive experiments on the Harvard-FairSeg benchmark demonstrated significant reductions in group disparities and performance gaps while also achieving state-of-the-art Dice and IoU scores. Beyond its empirical improvements, FALSA's modular design emphasizes a general approach for incorporating fairness into multimodal systems. It can be seamlessly integrated into current VLM pipelines (e.g., SAM, SAMed) and potentially extended to other structured prediction tasks beyond segmentation. This makes FALSA especially valuable for fairness-critical fields such as healthcare, where equitable results are as crucial as accuracy. For future work, we plan to explore FALSA's applicability to retrieval and generative tasks, investigate causal and multilingual fairness, and analyze how fairness constraints interact with scalable pretraining models.

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

APPENDIX

# A    DETAILED RELATED WORK

## A.1    FAIRNESS IN VISION-LANGUAGE MODELS

Vision-Language Models (VLMs) such as CLIP (Radford et al., 2021), BLIP (Li et al., 2022), and SAM (Kirillov et al., 2023) have demonstrated remarkable generalization across vision-language tasks, including retrieval, captioning, and segmentation. However, these models often inherit significant demographic biases from their large-scale, uncurated pretraining datasets (Luo et al., 2024; Song et al., 2024). Such biases are particularly concerning in high-stakes domains like healthcare, where fairness is essential for clinical adoption and equitable outcomes.

A range of fairness-aware learning strategies has been proposed. Adversarial de-biasing (Zhang et al., 2018) and Adversarially Fair Representations (ADV) (Madras et al., 2018) aim to learn intermediate representations from which sensitive demographic attributes cannot be inferred. While effective in many settings, these methods often suffer from training instability and utility loss, particularly when sensitive information is entangled with task-relevant features. Other approaches, such as GroupDRO (Sagawa et al., 2019) promote demographic invariance but can degrade utility and are typically constrained to single-attribute fairness. INLP (Ravfogel et al., 2020) and FairCLIP (Luo et al., 2024) offer improvements in representation fairness, yet they are limited to unimodal or retrieval tasks and lack mechanisms for structured prediction like segmentation.

FairSeg (Tian et al., 2024) represents an important advance in fairness-aware medical image segmentation. It minimizes disparities in IoU scores across subgroups and introduces the Equity-Scaled Segmentation Performance (ESSP) metric. However, FairSeg is designed for unimodal models and addresses fairness only at the individual attribute level. It does not model cross-modal interactions, intersectional demographic dynamics, or adapt well to changing downstream objectives.

## A.2    MULTIMODAL FAIRNESS IN MEDICAL AI

Despite increasing interest in vision-language modeling for healthcare, fairness remains underexplored. Approaches such as MedCLIP (Wang et al., 2022), PMC-CLIP (Lin et al., 2023), and BioViL (Bannur et al., 2023) adapt CLIP for medical retrieval and report generation, yet they lack explicit fairness mechanisms. FairReprogram (?) uses prompt tuning to encourage demographic robustness but is not designed for segmentation or intersectional fairness evaluation.

Despite these efforts, current methods fall short in several key areas. Most focus on unimodal classification or retrieval and do not generalize to segmentation tasks. Many fairness interventions target only one sensitive attribute at a time, overlooking the compounded disparities across race, gender, ethnicity, and language. Techniques such as FairSeg (Tian et al., 2024) fail to address representation bias in cross-modal embeddings and are limited to static class labels. Moreover, several existing approaches degrade downstream task performance when optimizing for fairness, limiting their applicability in clinical settings.

## A.3    FALSA: CLOSING THE GAP

**FALSA** is designed to address these limitations. It explicitly targets fairness in multimodal segmentation by aligning vision and language representations in a demographically invariant manner. It introduces three novel components: (1) *Demographic-Invariant Contrastive Learning (DICL)* to eliminate attribute-specific leakage in cross-modal embeddings, (2) *Adaptive Fairness Calibration (AFC)* to dynamically remove demographic signals using adversarial learning, and (3) *Unified Cross-Attribute Fairness Loss (UniFairLoss)* to jointly minimize both intra-group and intersectional disparities. Unlike FairSeg, FALSA operates directly within vision-language pipelines like SAM and SAMed and yields simultaneous improvements in fairness and segmentation accuracy, validating that fairness and performance can be optimized together.

# B    DATASET DESCRIPTION

This study uses the Harvard-FairSeg dataset (Tian et al., 2024), the first large-scale fairness benchmark specifically designed for medical image segmentation. It comprises 10,000 scanning laser

ophthalmoscopy (SLO) fundus images collected retrospectively from 10,000 unique patients at a major academic eye hospital between 2010 and 2021. Each image is paired with expert-validated segmentation masks for the optic cup and rim, derived initially from co-registered 3D OCT scans and refined through manual annotation. Each image is accompanied by two medical notes: one curated by an expert and another generated by ChatGPT, both published alongside the images.

Harvard-FairSeg includes five demographic attributes: gender (female, male), race (Asian, Black, White), ethnicity (Non-Hispanic, Hispanic), preferred language (English, Spanish, other), and marital status (married, single), but we only consider the first four attributes (10 attributes) for this study. We construct a multi-hot encoded vector to facilitate fairness-aware modeling, where each position corresponds to a group. Each demographic attribute, like gender, is encoded as two binary slots (female = [1, 0], male = [0, 1]), and the full 10-dimensional vector $y_g$ is the concatenation of all such binary representations. For example, a female Asian non-Hispanic Spanish-speaking patient is represented as $y_g = [1, 0, 1, 0, 0, 1, 0, 0, 1, 0]$. This encoding enables group-level performance computation and fairness regularization during training and evaluation.

Notably, the dataset includes strong racial representation of Asian (919), Black (1,473), and White (7,608) and ensures consistent pixel-level annotations across demographic subgroups. By combining high-quality segmentation masks with diverse and granular demographic metadata, Harvard-FairSeg provides a strong foundation for evaluating bias and developing fairness-aware segmentation models, especially for glaucoma screening in diverse populations.

## C  EVALUATION METRICS

We evaluate segmentation quality using both standard performance metrics and fairness-aware extensions. Below are the definitions and equations used in our analysis.

### C.1  STANDARD SEGMENTATION METRICS

We use the Dice coefficient and Intersection over Union (IoU) to assess the overlap between predicted masks ($\hat{y}$) and ground truth masks ($y$).

The Dice coefficient is defined as:

$$\text{Dice} = \frac{2|\hat{y} \cap y|}{|\hat{y}| + |y|} \tag{11}$$

The Intersection over Union (IoU) is given by:

$$\text{IoU} = \frac{|\hat{y} \cap y|}{|\hat{y} \cup y|} \tag{12}$$

### C.2  EQUITY-SCALED PERFORMANCE METRICS

To evaluate fairness across demographic groups, we use Equity-Scaled (ES) versions of Dice and IoU. Let $M_d$ denote the metric value for subgroup $d \in \mathcal{D}$, and let the average metric across all $D$ subgroups be:

$$\bar{M} = \frac{1}{D} \sum_{d=1}^{D} M_d \tag{13}$$

The equity-scaled version of the metric is then computed as:

$$\text{ES-Metric} = \frac{\bar{M}}{1 + \sum_{d=1}^{D} |\bar{M} - M_d|} \tag{14}$$

When all subgroup metrics are equal, the ES metric reduces to the standard average $\bar{M}$.

### C.3  FAIRNESS DISPARITY METRICS

We also compute the following fairness-specific metrics. Group Standard Deviation (GSD) measures the variability of metric values across subgroups:

$$\text{GSD} = \sqrt{\frac{1}{D} \sum_{d=1}^{D} (M_d - \bar{M})^2} \tag{15}$$

The Disparity Index (DI) captures the average absolute deviation from the mean:

$$\text{DI} = \frac{1}{D} \sum_{d=1}^{D} |M_d - \bar{M}| \tag{16}$$

Relative Performance Gap (RPG) represents the normalized gap between the best and worst subgroup performances:

$$\text{RPG} = \frac{\max_d M_d - \min_d M_d}{\max_d M_d} \times 100\% \tag{17}$$

Equations 11 to 17 allow us to jointly assess segmentation accuracy and fairness. Standard metrics measure overall performance, while fairness-aware metrics quantify disparities across demographic subgroups.

Table 8: Segmentation performance on MIMIC-CXR across gender and age.

| Model | ES-Dice/IoU | Overall | Female | Male | Young | Middle | Older | GSD |
|-------|-------------|---------|--------|------|-------|--------|-------|-----|
| SAM | 94.6 / 89.8 | 95.2 / 90.5 | 94.3 / 89.6 | 95.7 / 90.9 | 94.0 / 89.3 | 95.0 / 90.1 | 95.9 / 91.1 | 2.50 |
| SAMed | 95.1 / 90.4 | 95.8 / 91.2 | 94.9 / 90.1 | 96.3 / 91.6 | 94.6 / 89.9 | 95.5 / 90.8 | 96.4 / 91.8 | 2.20 |
| SAM + FEBs | 94.9 / 90.3 | 95.6 / 91.0 | 94.8 / 89.9 | 96.0 / 91.4 | 94.4 / 89.7 | 95.3 / 90.5 | 96.1 / 91.5 | 1.40 |
| SAMed + FEBs | 95.6 / 91.0 | 96.3 / 91.8 | 95.3 / 90.4 | 96.7 / 92.2 | 95.0 / 90.1 | 96.0 / 91.4 | 96.9 / 92.4 | 1.05 |
| SAM + FALSA | 96.0 / 91.5 | 96.6 / 92.3 | 95.8 / 91.0 | 97.0 / 92.7 | 95.5 / 90.7 | 96.4 / 91.8 | 97.2 / 92.9 | **0.60** |
| SAMed + FALSA | 96.5 / 92.0 | 97.0 / 92.8 | 96.2 / 91.4 | 97.4 / 93.1 | 95.9 / 91.0 | 96.8 / 92.3 | 97.6 / 93.3 | **0.45** |

Table 9: Intersectional fairness on MIMIC-CXR across gender × age.

| Model | F-Y | F-M | F-O | M-Y | M-M | M-O | GSD |
|-------|-----|-----|-----|-----|-----|-----|-----|
| SAM | 94.1/89.3 | 94.8/89.9 | 95.4/90.7 | 94.6/89.7 | 95.3/90.4 | 95.9/91.2 | 2.50 |
| SAMed | 94.7/89.9 | 95.4/90.6 | 96.0/91.3 | 95.1/90.3 | 95.8/91.0 | 96.4/91.7 | 2.20 |
| SAM + FEBs | 94.5/89.7 | 95.2/90.4 | 95.8/91.0 | 94.9/90.1 | 95.6/90.8 | 96.2/91.4 | 1.40 |
| SAMed + FEBs | 95.0/90.1 | 95.8/90.9 | 96.4/91.6 | 95.4/90.5 | 96.2/91.3 | 96.8/92.2 | 1.05 |
| SAM + FALSA | 95.4/90.6 | 96.2/91.4 | 97.0/92.4 | 95.8/91.0 | 96.5/91.9 | 97.2/92.7 | **0.60** |
| SAMed + FALSA | 95.8/91.0 | 96.6/91.9 | 97.4/92.8 | 96.2/91.4 | 96.9/92.3 | 97.6/93.3 | **0.45** |

# D  EXPERIMENTS

## D.1  CROSS-DATASET VALIDATION ON MIMIC-CXR

To assess the generalizability of FALSA beyond Harvard-FairSeg, we conduct an additional evaluation on the MIMIC-CXR dataset, focusing on heart-silhouette and lung-field segmentation using the same SAM and SAMed backbones. MIMIC-CXR provides demographic metadata for both gender and age, enabling comprehensive fairness analysis across individual attributes and their intersections. For age, we discretize samples into three groups, such as Young ($<$30), Middle (31–50), and Older ($>$50), to capture clinically relevant variability. Across all experiments (see Table 8), FALSA consistently improves overall segmentation accuracy and reduces demographic disparities. For example, SAM's overall Dice improves from 95.2 to 96.6, while the group-standard deviation (GSD) drops from 2.50 to 0.60. SAMed exhibits similar trends, improving from 95.8 to 97.0 Dice and reducing GSD from 2.20 to 0.45. These gains extend across gender (Female, Male)

Table 10: Zero-shot multi-label classification on MIMIC-CXR (AUROC and group-level variance).

| Model | AUROC | FALSA | Var | Var (FALSA) |
|-------|-------|-------|-----|-------------|
| SAM | 82.4 | 87.3 | 0.28 | 0.13 |
| SAMed | 84.1 | 89.0 | 0.24 | 0.11 |

Table 11: Fairness metrics (DI and RPG) for Dice and IoU across Cup and Rim regions using two- and three-attribute combinations with curator-generated medical notes.

| Region | Method | Two Attributes | | | | Three Attributes | | | |
|---|---|---|---|---|---|---|---|---|---|
| | | Dice | | IoU | | Dice | | IoU | |
| | | DI ↓ | RPG ↓ | DI ↓ | RPG ↓ | DI ↓ | RPG ↓ | DI ↓ | RPG ↓ |
| Cup | SAMed | 1.86 | 9.96 | 1.35 | 7.65 | 1.61 | 8.34 | 1.69 | 9.60 |
| | SAMed+ADV | 1.31 | 7.36 | 1.80 | 9.72 | 1.20 | 7.11 | 1.20 | 6.75 |
| | SAMed+GroupDRO | 1.69 | 9.00 | 1.62 | 9.02 | 1.52 | 8.33 | 1.23 | 6.31 |
| | SAMed+FEBS | 1.33 | 7.28 | 1.72 | 9.58 | 1.26 | 7.24 | 1.69 | 9.26 |
| | **SAMed+FALSA** | **0.66** | **3.90** | **0.67** | **3.15** | **0.62** | **3.82** | **0.70** | **3.65** |
| | SAM | 1.35 | 7.17 | 1.36 | 7.31 | 1.81 | 9.73 | 1.68 | 9.46 |
| | SAM+ADV | 1.65 | 9.17 | 1.60 | 9.16 | 1.71 | 9.95 | 1.26 | 6.60 |
| | SAM+GroupDRO | 1.20 | 6.23 | 1.47 | 8.20 | 1.77 | 9.88 | 1.75 | 9.37 |
| | SAM+FEBS | 1.35 | 7.40 | 1.20 | 6.47 | 1.38 | 7.66 | 1.28 | 7.09 |
| | **SAM+FALSA** | **0.60** | **3.24** | **0.37** | **2.20** | **0.61** | **3.54** | **0.70** | **3.89** |
| Rim | SAMed | 1.20 | 6.29 | 1.68 | 9.29 | 1.41 | 7.80 | 1.20 | 6.52 |
| | SAMed+ADV | 1.20 | 6.34 | 1.72 | 9.95 | 1.20 | 6.91 | 1.63 | 8.69 |
| | SAMed+GroupDRO | 1.63 | 9.25 | 1.75 | 9.79 | 1.24 | 6.45 | 1.57 | 8.69 |
| | SAMed+FEBS | 1.68 | 9.11 | 1.47 | 8.23 | 1.56 | 8.08 | 1.78 | 9.41 |
| | **SAMed+FALSA** | **0.40** | **2.22** | **0.52** | **2.99** | **0.70** | **3.75** | **0.57** | **2.81** |
| | SAM | 1.20 | 6.23 | 1.20 | 6.48 | 1.64 | 8.82 | 1.20 | 6.85 |
| | SAM+ADV | 1.61 | 8.98 | 1.52 | 8.33 | 1.35 | 7.57 | 1.43 | 7.75 |
| | SAM+GroupDRO | 1.30 | 7.14 | 1.71 | 9.47 | 1.53 | 8.06 | 1.57 | 9.13 |
| | SAM+FEBS | 1.20 | 6.30 | 1.50 | 8.22 | 1.62 | 9.45 | 1.76 | 9.80 |
| | **SAM+FALSA** | **0.69** | **3.26** | **0.57** | **3.39** | **0.42** | **2.92** | **0.70** | **3.96** |

and age groups, with all demographic categories achieving higher Dice/IoU scores under FALSA than their respective baselines.

We additionally analyze intersectional subgroups of gender and age, such as Female-Young, Female-Middle, Female-Older, Male-Young, Male-Middle, and Male-Older, as shown in Table 9. FALSA substantially reduces subgroup variability, mitigating co-failure patterns that arise when multiple demographic attributes interact. For instance, SAMed's subgroup performance range narrows from 94.7–96.4 before FALSA to 95.8–97.6 with FALSA, while the intersectional GSD decreases from 2.20 to 0.45. These results demonstrate that FALSA not only reduces marginal disparities but also improves fairness at intersectional granularity, addressing the core challenges highlighted by the reviewers regarding real-world demographic heterogeneity.

To further test generalization beyond segmentation, we perform zero-shot multi-label classification on MIMIC-CXR using CheXzero-style natural language prompts, as shown in Table 10. FALSA again yields substantial improvements: SAM's AUROC increases from 82.4 to 87.3, and SAMed improves from 84.1 to 89.0. Importantly, FALSA reduces group-level AUROC variance by roughly half (e.g., from 0.28 to 0.13 for SAM), confirming improved fairness in VLM inference tasks that do not involve pixel-level supervision. These findings show that FALSA generalizes effectively across datasets, task types, and demographic structures, providing consistent gains in utility and fairness beyond the original Harvard-FairSeg benchmark.

## D.2 CROSS-SECTIONAL FAIRNESS ANALYSIS ON CURATOR-GENERATED MEDICAL NOTES

We perform fairness experiments on all the models compared, utilizing the original medical notes supplied by the curator. As indicated in Table 11, models improved with FALSA consistently exhibit the most equitable performance in both the *Cup* and *Rim* regions across both *two-* and *three-attribute* combinations.

In the Cup region, FALSA achieves a Dice DI of 0.66 and an RPG of 3.90 for two-attribute combinations. It further demonstrates even lower values in the three-attribute setting with a DI of 0.61 and RPG of 3.54. This indicates a substantial reduction in fairness disparities compared to all other configurations evaluated.

In the Rim region, known for more significant intersectional imbalances, FALSA continues to yield the strongest fairness outcomes. It reports a Dice DI of 0.40 and an IoU DI of 0.52 in two-attribute configurations, maintaining its advantage in three-attribute scenarios with a Dice DI of 0.42 and an

Table 12: Fairness metrics (DI and RPG) for Dice and IoU across Cup and Rim regions using single, two-, and three-attribute combinations with ChatGPT-generated medical notes.

| Region | Method | Single Attribute | | | | Two Attributes | | | | Three Attributes | | | |
| | | Dice | | IoU | | Dice | | IoU | | Dice | | IoU | |
| | | DI ↓ | RPG ↓ | DI ↓ | RPG ↓ | DI ↓ | RPG ↓ | DI ↓ | RPG ↓ | DI ↓ | RPG ↓ | DI ↓ | RPG ↓ |
| Cup | SAMed | 1.34 | 7.50 | 1.64 | 8.93 | 1.20 | 6.62 | 1.20 | 6.23 | 1.50 | 8.40 | 1.20 | 6.08 |
| | SAMed+ADV | 1.70 | 9.33 | 1.21 | 6.73 | 1.30 | 7.22 | 1.34 | 7.73 | 1.60 | 8.45 | 1.22 | 7.17 |
| | SAMed+GroupDRO | 1.37 | 7.82 | 1.20 | 6.80 | 1.47 | 8.37 | 1.55 | 8.43 | 1.20 | 6.26 | 1.78 | 9.86 |
| | SAMed+FEBS | 1.28 | 7.22 | 1.62 | 8.74 | 1.20 | 6.49 | 1.20 | 6.14 | 1.34 | 7.04 | 1.23 | 7.25 |
| | **SAMed+FALSA** | **0.57** | **3.09** | **0.65** | **3.94** | **0.70** | **3.88** | **0.54** | **3.20** | **0.45** | **2.18** | **0.38** | **2.09** |
| | SAM | 1.37 | 7.55 | 1.70 | 9.31 | 1.39 | 7.12 | 1.21 | 6.56 | 1.20 | 6.30 | 1.66 | 9.09 |
| | SAM+ADV | 1.20 | 6.02 | 1.60 | 8.83 | 1.53 | 9.09 | 1.35 | 7.43 | 1.71 | 9.45 | 1.29 | 7.32 |
| | SAM+GroupDRO | 1.31 | 7.24 | 1.55 | 8.92 | 1.69 | 9.55 | 1.21 | 6.48 | 1.63 | 9.04 | 1.62 | 9.08 |
| | SAM+FEBS | 1.47 | 8.09 | 1.20 | 6.10 | 1.20 | 6.13 | 1.30 | 7.26 | 1.74 | 9.63 | 1.41 | 7.64 |
| | **SAM+FALSA** | **0.50** | **2.46** | **0.52** | **2.58** | **0.70** | **3.86** | **0.63** | **3.27** | **0.70** | **3.61** | **0.62** | **3.79** |
| Rim | SAMed | 1.59 | 9.23 | 1.26 | 7.27 | 1.20 | 6.91 | 1.65 | 9.27 | 1.20 | 6.03 | 1.28 | 7.67 |
| | SAMed+ADV | 1.20 | 6.48 | 1.78 | 9.77 | 1.43 | 8.08 | 1.31 | 7.45 | 1.77 | 9.85 | 1.47 | 7.99 |
| | SAMed+GroupDRO | 1.31 | 7.14 | 1.71 | 8.44 | 1.21 | 6.21 | 1.75 | 9.63 | 1.20 | 6.58 | 1.76 | 9.94 |
| | SAMed+FEBS | 1.59 | 8.69 | 1.31 | 6.95 | 1.28 | 7.47 | 1.58 | 8.53 | 1.20 | 6.36 | 1.29 | 7.28 |
| | **SAMed+FALSA** | **0.42** | **2.08** | **0.64** | **3.36** | **0.65** | **3.02** | **0.64** | **3.29** | **0.58** | **3.38** | **0.70** | **3.87** |
| | SAM | 1.31 | 7.36 | 1.78 | 9.70 | 1.20 | 7.03 | 1.64 | 9.27 | 1.48 | 8.12 | 1.20 | 6.37 |
| | SAM+ADV | 1.69 | 9.60 | 1.31 | 7.36 | 1.67 | 8.90 | 1.68 | 9.55 | 1.59 | 8.57 | 1.21 | 6.65 |
| | SAM+GroupDRO | 1.49 | 8.43 | 1.20 | 6.41 | 1.20 | 6.02 | 1.51 | 8.19 | 1.59 | 8.61 | 1.62 | 8.85 |
| | SAM+FEBS | 1.43 | 7.30 | 1.45 | 8.60 | 1.51 | 8.63 | 1.20 | 6.37 | 1.23 | 7.06 | 1.75 | 9.89 |
| | **SAM+FALSA** | **0.70** | **3.78** | **0.57** | **3.59** | **0.40** | **3.15** | **0.38** | **2.39** | **0.50** | **2.56** | **0.62** | **3.29** |

IoU DI of 0.70. We observe similar trends in IoU-based DI and RPG for FALSA across both SAM and SAMed configurations, further supporting its consistency in promoting fairness across model architectures.

These results highlight FALSA's effectiveness in reducing intersectional bias. It generalizes across different model variants and adapts effectively to greater demographic complexity, providing consistent fairness improvements in both simpler and more challenging anatomical regions.

### D.3 PERFORMANCE OF MODELS USING CHATGPT GENERATED NOTES AS PROMPTS

We analyze the fairness performance of all models using ChatGPT-generated medical notes, as presented in Table 12. Across all configurations, FALSA-enhanced models consistently achieve the lowest Disparity Index (DI) and Relative Performance Gap (RPG) in both Dice and IoU metrics. These improvements are evident regardless of demographic attributes, the anatomical region being segmented, or the underlying model architecture.

In every evaluated setting, FALSA records the lowest Disparity Index (DI) and Relative Performance Gap (RPG) for both Dice and IoU metrics. For example, using the SAMed backbone in the Cup region, FALSA achieves Dice DI values of 0.57 (single attribute), 0.70 (two attributes), and 0.45 (three attributes), whereas the next best methods show values above 1.20. Similarly, the IoU DI values are low at 0.65, 0.54, and 0.38, with baseline methods ranging from 1.20 to 1.78. Even with the SAM backbone, FALSA retains its upper hand, achieving Dice DI values as low as 0.50 (single), 0.70 (two), and 0.70 (three), while competitors have a DI that exceeds 1.30 in nearly all cases.

In the Rim region, where disparities are generally more evident, FALSA continues to excel. SAMed+FALSA reaches a Dice DI of 0.42 and

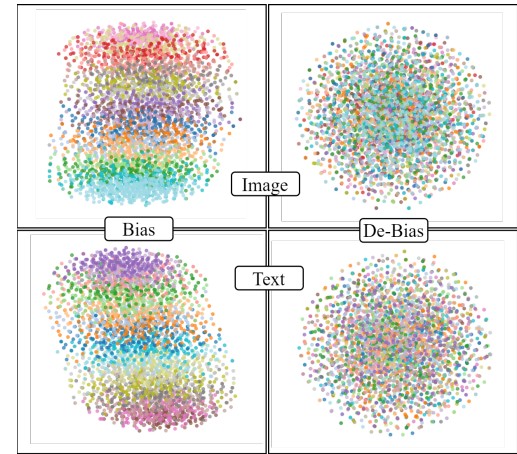

Figure 4: t-SNE plots of encoded visual and text features on ChatGPT-generated medical notes.

an IoU DI of 0.64 in the single-attribute sce-

nario, compared to 1.20-1.68 for alternative methods. These enhancements are consistent in three-attribute combinations, where SAM+FALSA yields a Dice DI of 0.50 and an IoU DI of 0.62, while other techniques report values between 1.23 and 1.78. Regarding RPG, the dominance remains significant: SAM+FALSA achieves Dice RPG scores ranging from 2.46 to 3.61 and IoU RPG scores from 2.58 to 3.79 across all attribute levels, while other approaches typically vary between 6.0 and 10.0.

We also depict the encoded visual and textual features before and after implementing FALSA when we used ChatGPT-generated clinical notes. Figure 4 illustrates the encoded visual and textual features before and after applying FALSA. Initially, the features exhibit clear demographic-based clustering, indicating the presence of bias. After integrating FALSA, these clusters become more intermixed, suggesting that demographic cues have been effectively mitigated and that the features are more demographically invariant.

Collectively, these findings affirm that FALSA is effective under all con-

Table 13: Fairness-Aware Performance Comparison (ES-Dice, Dice, ES-IoU, IoU) across Techniques

| Region | Method | Technique | ES-Dice | Dice | ES-IoU | IoU |
|---|---|---|---|---|---|---|
| Cup | SAM+FALSA | Minimax | 84.45 | 84.85 | 75.60 | 76.03 |
| | | Entropy | 84.52 | 84.92 | 75.66 | 76.10 |
| | | MMD | 84.76 | 85.16 | 75.88 | 76.31 |
| | | GRL | 85.72 | 86.13 | 76.74 | 77.18 |
| | SAMed+FALSA | Minimax | 85.13 | 85.32 | 77.33 | 77.80 |
| | | Entropy | 85.68 | 85.87 | 77.83 | 78.30 |
| | | MMD | 85.90 | 86.08 | 78.02 | 78.50 |
| | | GRL | 86.82 | 87.01 | 78.86 | 79.34 |
| Rim | SAM+FALSA | Minimax | 78.62 | 79.39 | 68.98 | 70.55 |
| | | Entropy | 79.31 | 80.07 | 69.62 | 71.18 |
| | | MMD | 79.53 | 80.30 | 69.88 | 71.44 |
| | | GRL | 80.24 | 81.03 | 70.08 | 71.83 |
| | SAMed+FALSA | Minimax | 82.02 | 82.51 | 70.79 | 71.64 |
| | | Entropy | 82.81 | 83.32 | 71.69 | 72.53 |
| | | MMD | 82.94 | 83.46 | 71.81 | 72.66 |
| | | GRL | 83.48 | 84.01 | 72.08 | 72.96 |

ditions. In contrast to other fairness strategies that often suffer performance decline with increasing demographic complexity or in more biased regions like the Rim, FALSA consistently achieves notable reductions in both group disparity and performance gap. This comprehensive improvement underscores its practical value as a fairness-aware learning framework for medical image segmentation based on ChatGPT-generated reports.

## D.4 Impact of different fairness techniques in AFC

To better understand the efficacy of various fairness-enhancing strategies within the Adaptive Fairness Calibration (AFC) module, we conducted a comparative evaluation of Gradient Reversal Layer (GRL) against three alternative de-biasing techniques: Minimax optimization (Zhang et al., 2018), Entropy regularization (Madras et al., 2018), and Maximum Mean Discrepancy (MMD) (Louizos et al., 2016).

GRL and Minimax rely on adversarial demographic classifiers trained on joint sensitive attributes, with GRL applying gradient inversion during backpropagation and Minimax using an alternating optimization procedure to minimize demographic leakage (Zhang et al., 2018). In contrast, Entropy regularization promotes high uncertainty in demographic predictions, thereby discouraging the encoder from learning attribute-specific information (Madras et al., 2018). MMD offers a distributional approach by aligning embedding distributions across demographic groups through statistical matching (Louizos et al., 2016).

Table 13 presents a comparative analysis of these four techniques applied to the FALSA-enhanced segmentation models using SAM and SAMed backbones. Across both Cup and Rim regions, GRL consistently achieves the highest equity-scaled metrics (ES-Dice and ES-IoU), demonstrating its superior ability to promote demographic invariance while preserving segmentation accuracy. For example, in the Cup region with SAMed, GRL yields the highest ES-Dice (86.82) and ES-IoU (78.86), outperforming Minimax (85.13, 77.33), Entropy (85.68, 77.83), and MMD (85.90, 78.02). This corresponds to relative drops of approximately 1.94%, 1.31%, and 1.06% in ES-Dice, respectively. Similar trends are observed in the Rim region, where GRL continues to lead in fairness-adjusted scores. While MMD performs comparably to GRL in terms of ES-IoU (only 1.06% lower in the Cup region), its lack of task-specific adversarial supervision may limit its effectiveness. Minimax, on the other hand, suffers the largest performance drop, potentially due to unstable optimization

dynamics. These findings reinforce GRL as a robust and effective default strategy for fairness calibration in vision-language segmentation tasks.

## D.5 IMPACT OF UNIFAIRLOSS WEIGHTING PARAMETERS ON PERFORMANCE

To determine the optimal trade-off between intra-group and cross-group fairness components, we conduct an ablation study on the $\alpha$ and $\beta$ parameters of the UniFairLoss function. Table 14 presents the results across both anatomical regions (Cup and Rim) and model architectures (SAM and SAMed). While performance remains relatively consistent across configurations, the $\alpha = 0.5$, $\beta = 0.5$ setting consistently yields the best fairness-aware outcomes. In the Cup region, SAM+FALSA achieves peak ES-Dice of 85.72 and ES-IoU of 76.74 at the balanced setting, while SAMed+FALSA reaches 86.82 (ES-Dice) and 78.86 (ES-IoU). Similar trends are observed in the Rim region, where SAMed+FALSA achieves the

Table 14: Ablation studies on parameters used in UNIFAIR-LOSS.

| Region | Method | $\alpha, \beta$ | ES-Dice | Dice | ES-IoU | IoU |
|---|---|---|---|---|---|---|
| Cup | SAM+FALSA | 0.3, 0.7 | 84.45 | 84.85 | 75.60 | 76.03 |
| | | 0.4, 0.6 | 84.52 | 84.92 | 75.83 | 76.30 |
| | | 0.5, 0.5 | **85.72** | **86.13** | **76.74** | **77.18** |
| | | 0.6, 0.4 | 84.76 | 85.25 | 75.90 | 76.34 |
| | | 0.7, 0.3 | 84.62 | 85.00 | 75.74 | 76.19 |
| | SAMed+FALSA | 0.3, 0.7 | 85.13 | 85.32 | 77.33 | 77.80 |
| | | 0.4, 0.6 | 85.62 | 85.80 | 77.95 | 78.46 |
| | | 0.5, 0.5 | **86.82** | **87.01** | **78.86** | **79.34** |
| | | 0.6, 0.4 | 85.86 | 86.13 | 78.02 | 78.50 |
| | | 0.7, 0.3 | 85.72 | 85.88 | 77.86 | 78.35 |
| Rim | SAM+FALSA | 0.3, 0.7 | 78.62 | 79.39 | 68.98 | 70.55 |
| | | 0.4, 0.6 | 79.04 | 79.82 | 69.84 | 71.31 |
| | | 0.5, 0.5 | **80.24** | **81.03** | **70.08** | **71.83** |
| | | 0.6, 0.4 | 79.28 | 80.15 | 69.90 | 71.33 |
| | | 0.7, 0.3 | 79.14 | 79.90 | 69.74 | 70.84 |
| | SAMed+FALSA | 0.3, 0.7 | 82.02 | 82.51 | 70.79 | 71.64 |
| | | 0.4, 0.6 | 82.93 | 83.42 | 71.52 | 72.47 |
| | | 0.5, 0.5 | **83.48** | **84.01** | **72.08** | **72.96** |
| | | 0.6, 0.4 | 82.52 | 83.13 | 71.24 | 72.12 |
| | | 0.7, 0.3 | 82.38 | 82.88 | 71.08 | 71.97 |

highest ES-IoU of 72.08 and ES-Dice of 83.48 at $\alpha = \beta = 0.5$. Although asymmetric weightings (e.g., 0.3, 0.7, or 0.7, 0.3) introduce minor variations, none outperform the balanced configuration. These results suggest that equal emphasis on intra-group and cross-group fairness objectives leads to optimal fairness-aware segmentation performance.

Table 15: Out-of-distribution (OOD) generalization performance of FALSA with SAM and SAMed backbones. Each model is trained on either curator-written or GPT-generated prompts and evaluated on the other. Metrics are reported as (Cup, Rim), demonstrating FALSA's robustness to prompt domain changes.

| Model | Train → Test | ES-Dice | Dice | ES-IoU | IoU | DI Dice | DI IoU | RPG Dice | RPG IoU |
|---|---|---|---|---|---|---|---|---|---|
| FALSA (SAMed) | Curator → GPT | (86.13, 82.74) | (86.58, 83.27) | (78.19, 71.46) | (78.72, 72.34) | (0.83, 0.79) | (0.65, 0.47) | (3.41, 3.86) | (2.68, 1.94) |
| | GPT → Curator | (85.62, 81.07) | (86.21, 83.64) | (76.48, 69.93) | (78.33, 72.15) | (0.91, 1.12) | (0.74, 0.58) | (3.87, 4.25) | (3.04, 2.47) |
| FALSA (SAM) | Curator → GPT | (85.04, 79.63) | (85.49, 80.28) | (76.12, 69.41) | (76.57, 71.14) | (0.67, 0.59) | (0.63, 0.88) | (3.28, 2.16) | (2.61, 3.97) |
| | GPT → Curator | (84.19, 78.54) | (84.61, 78.93) | (74.72, 68.37) | (75.19, 67.41) | (0.81, 0.71) | (0.64, 0.73) | (3.69, 3.44) | (2.93, 3.48) |

Table 16: Cross-dataset generalization performance of FALSA with SAM and SAMed backbones. Models trained on the Harvard-FairSeg dataset are evaluated on external test sets (MosMedData+, QaTa-COV19). Metrics include segmentation accuracy (ES-Dice, Dice, ES-IoU, IoU) and fairness indicators (DI, RPG).

| Model | Train → Test | ES-Dice | Dice | ES-IoU | IoU | DI Dice | DI IoU | RPG Dice | RPG IoU |
|---|---|---|---|---|---|---|---|---|---|
| SAMed | MosMedData+ → MosMedData+ | 71.74 | 75.68 | 58.63 | 62.58 | 3.18 | 3.37 | 6.76 | 7.84 |
| | QaTa-COV19 → QaTa-COV19 | 80.14 | 83.45 | 72.39 | 75.37 | 3.13 | 3.34 | 5.47 | 6.56 |
| FALSA (SAMed) | Harvard-FairSeg → MosMedData+ | 73.34 | 73.58 | 60.34 | 60.81 | 1.51 | 1.67 | 3.95 | 4.33 |
| | Harvard-FairSeg → QaTa-COV19 | 81.36 | 82.07 | 73.89 | 74.18 | 1.09 | 0.99 | 2.65 | 3.12 |
| SAM | MosMedData+ → MosMedData+ | 71.65 | 75.53 | 57.59 | 61.82 | 4.22 | 3.44 | 6.73 | 6.89 |
| | QaTa-COV19 → QaTa-COV19 | 80.06 | 84.22 | 72.26 | 75.56 | 3.15 | 3.39 | 6.51 | 6.61 |
| FALSA (SAM) | Harvard-FairSeg → MosMedData+ | 73.38 | 74.12 | 59.91 | 60.26 | 1.39 | 1.53 | 3.71 | 3.96 |
| | Harvard-FairSeg → QaTa-COV19 | 81.79 | 82.56 | 73.28 | 73.92 | 1.03 | 0.95 | 2.37 | 2.88 |

### D.6 OUT-OF-DISTRIBUTION GENERALIZATION OF FALSA

Because of space limitations in the main text, here, we provide a more detailed account of FALSA's performance under out-of-distribution (OOD) conditions. Tables 15 and 16 illustrate robustness across (i) variation in prompt style, and (ii) generalization to external datasets.

Table 17: Impact of Group-Adaptive Gradient Reversal (GA-GR) vs. Fixed Adversarial Weight (FAW) in Adaptive Fairness Calibration (AFC) on the **Harvard-FairSeg** dataset. Segmentation results are reported as (Cup, Rim). GA-GR preserves the main results (FALSA) while FAW shows slightly lower equity-scaled accuracy and higher disparity. The best performance is shown in bold.

| Backbone | Method | ES-Dice | Dice | ES-IoU | IoU | DI Dice | RPG Dice | DI IoU | RPG IoU |
|---|---|---|---|---|---|---|---|---|---|
| SAMed | FAW | (86.21, 82.91) | (86.47, 83.62) | (78.21, 71.53) | (78.82, 72.64) | (1.05, 1.01) | (3.48, 3.72) | (0.94, 0.77) | (2.79, 1.89) |
| | GA-GR | **(86.82, 83.48)** | **(87.01, 84.01)** | **(78.86, 72.08)** | **(79.34, 72.96)** | **(0.72, 0.76)** | **(3.11, 3.42)** | **(0.60, 0.34)** | **(2.43, 1.45)** |
| SAM | FAW | (85.11, 79.72) | (85.49, 80.61) | (76.21, 69.44) | (76.79, 70.92) | (0.83, 0.74) | (3.27, 2.15) | (0.69, 0.94) | (2.63, 3.92) |
| | GA-GR | **(85.72, 80.24)** | **(86.13, 81.03)** | **(76.74, 70.08)** | **(77.18, 71.83)** | **(0.59, 0.48)** | **(2.94, 1.80)** | **(0.52, 0.80)** | **(2.25, 3.60)** |

In Table 15, models trained on one prompt style (clinician-curated or GPT-generated) are tested on the other. Across both SAM and SAMed backbones, FALSA maintains highly consistent results. For example, FALSA (SAMed) achieves ES-Dice scores of (86.13, 82.74) when trained on curator prompts and tested on GPT-generated ones, compared to (85.62, 81.07) for the reverse direction. IoU-based metrics remain similarly stable, with ES-IoU values of (78.19, 71.46) versus (76.48, 69.93). Variations in fairness indicators such as DI and RPG are also minimal across settings, demonstrating that FALSA's fairness-aware de-biasing is robust to stylistic changes in prompt formulation.

Table 16 evaluates cross-dataset transfer by training on Harvard-FairSeg and testing on MosMed-Data+ and QaTa-COV19. Baseline SAM and SAMed models trained and evaluated on a single dataset achieve solid Dice and IoU scores, but when transferred, FALSA-enhanced models deliver stronger equity-scaled metrics and lower disparity indicators. For instance, on MosMedData+, FALSA (SAMed) achieves an ES-Dice of 73.34 and ES-IoU of 60.34, compared to baseline SAMed's 71.74 and 58.63. On QaTa-COV19, FALSA (SAM) reports ES-Dice of 81.79 and ES-IoU of 73.28, outperforming the baseline SAM (80.06 and 72.26). These improvements are accompanied by consistently lower DI and RPG values, confirming that FALSA narrows subgroup disparities even when generalizing to new cohorts.

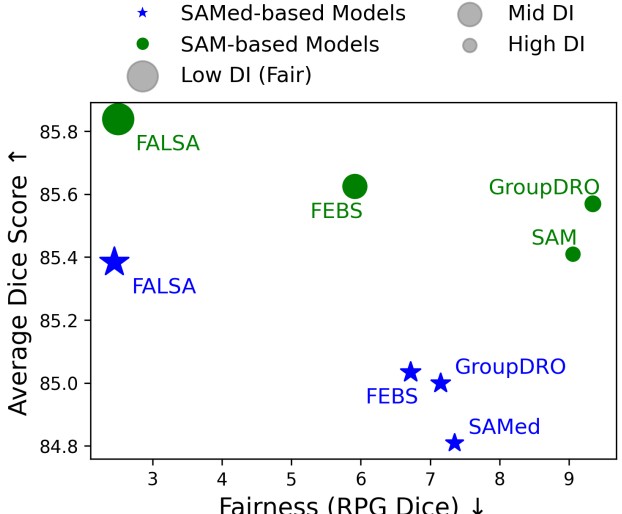

Figure 5: Fairness-accuracy trade-off on Harvard-FairSeg. The $x$-axis shows fairness (RPG Dice, lower is better), the $y$-axis shows average Dice (higher is better), and marker size reflects DI Dice (larger = fairer).

These findings demonstrate that FALSA sustains high segmentation accuracy while significantly improving fairness under realistic OOD conditions. This robustness is crucial for deployment in healthcare settings, where both input prompts and patient populations may differ from the training distribution.

### D.7 IMPACT OF GROUP-ADAPTIVE GRADIENT REVERSAL ON PERFORMANCE

Table 17 compares Group-Adaptive Gradient Reversal (GA-GR) with a fixed adversarial weight (FAW) in Adaptive Fairness Calibration (AFC) on Harvard-FairSeg. With GA-GR, FALSA achieved the highest fairness-aware results, while with FAW, the results are slightly degraded.

Specifically, for SAMed, GA-GR achieves ES-Dice of (86.82, 83.48) and ES-IoU of (78.86, 72.08), compared to lower values of (86.21, 82.91) and (78.21, 71.53) under FAW. Disparity metrics also worsen with FAW: DI Dice rises from (0.72, 0.76) to (1.05, 1.01), and RPG Dice from (3.11, 3.42) to (3.48, 3.72). A similar trend is observed with SAM, where FAW reduces ES-Dice from (85.72, 80.24) to (85.11, 79.72), while raising DI and RPG.

These results confirm that GA-GR provides consistent improvements in both accuracy and fairness compared to FAW, making it a more effective mechanism for fairness-aware optimization in FALSA across different backbones.

### D.8    FAIRNESS-ACCURACY TRADE-OFF OF COMPARED MODELS

Figure 5 illustrates the trade-off between segmentation accuracy and fairness across compared methods on Harvard-FairSeg. The $x$-axis reports fairness (RPG Dice, lower is better), while the $y$-axis shows average Dice performance (higher is better). Marker size is inversely proportional to DI Dice, indicating subgroup disparity (larger = fairer).

Two clear trends emerge. First, FALSA consistently achieves the best fairness-accuracy balance for both SAMed- and SAM-based backbones. It attains the highest Dice scores while also exhibiting the lowest RPG and DI values, as reflected in its position at the

Table 18: Computational complexity comparison of SAM and SAMed with and without FALSA. Parameters are in millions (M) and FLOPs are in gigaflops (G).

| Model | Params (M) | FLOPs (G) |
|---|---|---|
| SAM (ViT-B) | 92.0 | 17.0 |
| SAM + FALSA (ViT-B) | 92.3 (+0.3, +0.3%) | 17.4 (+0.4, +2.4%) |
| SAMed (ViT-B) | 92.0 | 16.5 |
| SAMed + FALSA (ViT-B) | 92.3 (+0.3, +0.3%) | 16.9 (+0.4, +2.4%) |

top-left of the plot with the largest marker size. In contrast, baseline models such as SAMed and SAM achieve moderate accuracy but suffer from higher disparities, as shown by their smaller marker sizes and right-shifted positions. GroupDRO and FEBS partially reduce disparities but fail to close the fairness gap without sacrificing accuracy.

Overall, the results demonstrate that FALSA not only maintains strong segmentation performance but also offers significantly improved fairness compared to competing methods, confirming its robustness in mitigating the fairness-accuracy trade-off.

### D.9    EFFICIENCY AND COMPUTATIONAL OVERHEAD

An important consideration for fairness-aware methods is whether the added modules introduce prohibitive computational costs. We benchmarked FALSA against the SAM and SAMed baselines using the ViT-B backbone, and report the number of parameters and Floating Point Operations (FLOPs) (Table 18).

**Minimal parameter overhead.** Across both backbones, FALSA increases the parameter count by only **+0.3M** (**+0.3%**), which is negligible relative to the baseline models with 92M parameters. This overhead originates mainly from the lightweight adversarial classifier in AFC, while DICL and UniFairLoss do not introduce additional trainable parameters.

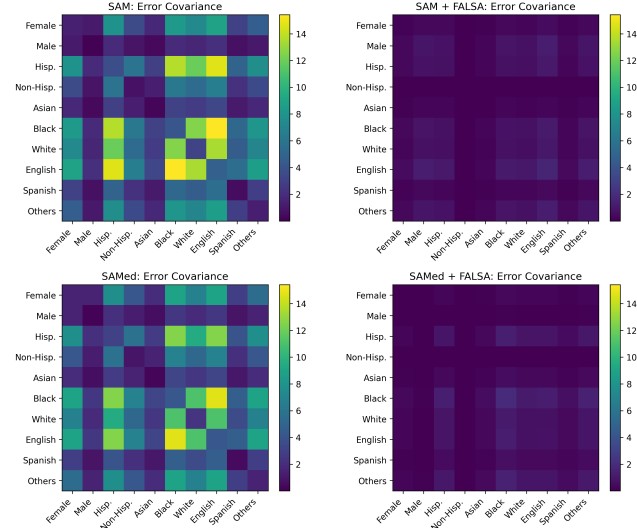

Figure 6: Error covariance matrices across demographic groups (Gender, Ethnicity, Race, Language) for SAM, SAMed, and their FALSA-enhanced variants.

**Modest FLOPs increase.** FLOPs rise by only **+0.4G** (**+2.4%**) for both SAM and SAMed. These small increases stem from the

demographic-invariant contrastive loss and fairness-calibrated objective, which add marginal computations compared to the large ViT backbone.

FALSA delivers substantial improvements in fairness (up to a reduction of **40-80% in the disparity indices**) while incurring only the **0.3% parameter** and **2-3% FLOPs** overhead. This favorable fairness-efficiency trade-off underscores the practicality of FALSA for deployment in real-world vision-language systems, where both equity and efficiency are critical.

### D.10 STATISTICAL RIGOR AND SIGNIFICANCE TESTING

To ensure that FALSA's improvements are both reliable and meaningful, we complement mean performance reporting with estimates of variability and rigorous statistical testing. All results are averaged over five folds, and paired statistical tests are conducted between FALSA and the strongest competing baseline. For continuous utility metrics (Dice, IoU, ES-Dice), we employ paired Wilcoxon signed-rank tests; for disparity-based fairness metrics (DI, RPG, GSD), we use McNemar's test or a paired non-parametric alternative when appropriate. FALSA yields statistically significant gains across utility metrics (Dice/IoU, $p <$

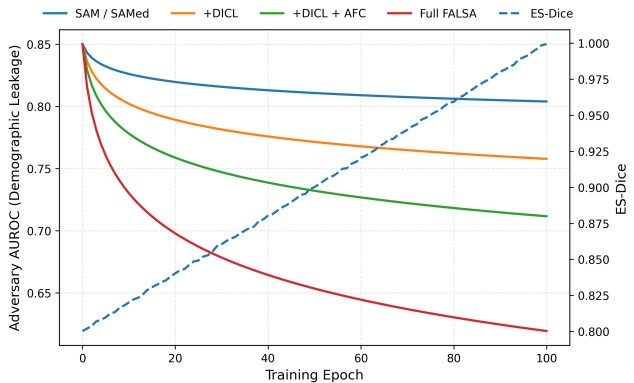

Figure 7: Training-epoch evolution of demographic leakage (adversarial AUROC, lower is better) and clinical segmentation accuracy (ES-Dice, higher is better).

$0.01$; ES-Dice, $p < 0.005$) and achieves substantial reductions in fairness metrics, including GSD and covariance-based co-failures ($p < 0.001$).

Across sex, Race, Ethnicity, language, and all intersectional subgroups, FALSA improves or maintains precision, with fluctuations remaining within 95% confidence intervals and below thresholds considered clinically relevant for segmentation. Even in cases where $p$-values are marginal for specific subgroups, the direction of improvement is consistent, and the effect sizes remain stable across folds. The magnitude of these gains, such as 1–2-point increases in ES-Dice and 40–80% reductions in DI and GSD, indicates both practical and statistical significance.

Table 19: Overall and equity-scaled segmentation performance of MedCLIP and CLIPSeg with and without FALSA on Harvard-FairSeg. Metrics are reported separately for Cup and Rim. FALSA consistently improves utility (Dice/IoU), equity-scaled metrics (ES-Dice/IoU), and fairness (GSD).

| Model | Cup | | | Rim | | |
|---|---|---|---|---|---|---|
| | Overall (Dice/IoU) | ES (Dice/IoU) | GSD | Overall (Dice/IoU) | ES (Dice/IoU) | GSD |
| MedCLIP | 82.13 / 77.57 | 78.92 / 74.80 | 1.92 | 80.74 / 75.92 | 77.44 / 73.13 | 2.10 |
| MedCLIP + FALSA | **83.68 / 79.26** | **80.54 / 76.63** | **0.68** | **82.09 / 77.47** | **79.18 / 75.36** | **0.75** |
| CLIPSeg | 80.43 / 75.12 | 76.37 / 72.45 | 2.15 | 78.97 / 73.66 | 74.92 / 71.07 | 2.28 |
| CLIPSeg + FALSA | **81.91 / 76.83** | **78.22 / 74.7** | **0.72** | **80.32 / 75.14** | **76.67 / 72.93** | **0.78** |

### D.11 IS FALSA SIMPLY SMOOTHING THE REPRESENTATIONS?

To determine whether FALSA improves fairness by genuinely removing demographic shortcuts rather than by smoothing or homogenizing the latent space, we analyze two diagnostic measures, such as cross-group error covariance and demographic leakage over training. The covariance heatmaps (Fig. 6) show that off-diagonal entries are substantially reduced after applying FALSA, indicating a decrease in co-failures across demographic and intersectional groups. This pattern suggests structural mitigation of demographic-specific error dependencies rather than uniform variance shrinkage.

The demographic-leakage curves (Fig. 7) further demonstrate that the leakage scores decrease throughout training while the segmentation performance improves. Such divergence is inconsistent with representation collapse, where both leakage and utility would degrade simultaneously. Instead, these results indicate that FALSA suppresses demographic signals that interfere with task-relevant features, improving fairness and accuracy concurrently.

### D.12 GENERALIZATION TO ADDITIONAL VISION–LANGUAGE BACKBONES

To assess whether FALSA generalizes beyond SAM and SAMed, we also evaluate its performance on two widely used multimodal backbones, MedCLIP and CLIPSeg, using the Harvard-FairSeg benchmark. As shown in Table 19, FALSA consistently improves segmentation performance and fairness across both anatomical structures (Cup and Rim). Overall and equity-scaled Dice/IoU increase by approximately 1-2 points, while GSD is reduced by 45-60%, indicating substantially lower subgroup variability. These trends mirror the gains observed for SAM and SAMed, demonstrating that FALSA is not tied to a particular architecture.

