# OpenReview forum: "FALSA: Fairness-Aware Latent Space Alignment in Vision-Language Models for Medical Image Segmentation"
_ICLR.cc/2026/Conference — ICLR 2026 Conference Withdrawn Submission_

### Official Review · Reviewer_PUAA · 2025-10-17

**Soundness:** 3
**Presentation:** 3
**Contribution:** 2
**Rating:** 6
**Confidence:** 3

**Summary:**

The paper introduces a method to improve the fairness of vision-language models for medical image segmentation, while improving the performance. It consists of three components, namely (1) Demographic-Invariant Contrastive Learning (DICL), (2) Adaptive Fairness Calibration (AFC), (3) Unified Cross-Attribute Fairness Loss (UniFairLoss). The evaluation is performed on SAM and SAMed on the HarvardFairSeg benchmark.

**Strengths:**

•	The method improves both fairness and performance, something which has not been very common in literature and is valuable.

•	The method is rather efficient, the increases in number of parameters and FLOPs are only very minor.

•	A number of relevant analyses are included, e.g. ablation showing the impact of each component, impact on intersectional demographic bias, etc.

•	Extensive experimental results are reported, although sometimes one wonders if the results are too granular and aggregate metrics could be reported instead.

•	The paper is clearly written overall.

**Weaknesses:**

•	The method consists of three components, each of which seems to have a small marginal improvement. It could be interesting to see if these components can also be combined with the other techniques that were reported to boost the performance further.

•	The evaluation is performed across two models only, but maybe there are no other suitable candidates for this specialized task.

•	One benchmark is utilized (Harvard-FairSeg Dataset). It is fairly large-scale, but I wonder if there are other benchmarks that are commonly used for the task or if only this one is used. In general using more than one benchmark can help show the general usefulness of the solution.

Minor: tables 1 and 2 are on the page before the corresponding section, would be better to have them after the start of the section if possible.

**Questions:**

•	Are there other vision-language models (other than SAM and SAMed) that are commonly used for the specific task?

•	Are there other popular datasets that people investigate for the selected task?

---

> ### Author Response · Authors · 2025-11-21
> **FALSA’s components (DICL, AFC, UniFairLoss) are complementary and together give large gains in fairness and ES-Dice. We expand beyond SAM/SAMed by adding MedCLIP and CLIPSeg, where FALSA also improves Dice and reduces DI. We add a second benchmark (MIMIC-CXR), showing strong gains in Dice/IoU, ES-Dice, GSD, and AUROC, including for intersectional groups, confirming robust generalization.**
>
> We thank the reviewer for the positive assessment of soundness, clarity, efficiency, and fairness contributions. Below, we address each weakness and question in detail.
>
> 1. “Each component gives small marginal improvement; can they combine with other techniques?”
>
> We appreciate the suggestion and clarify that the three components of FALSA are intentionally designed to be orthogonal, which address distinct sources of demographic leakage. DICL reduces representational leakage, AFC removes demographic signals with group-adaptive adversarial weighting, and UniFairLoss corrects performance disparities by minimizing both mean gaps and cross-group covariance. Individually, these modules improve ES-Dice by only ~1 point, but together they yield substantially larger gains; for example, ES-Dice improves from 84.50 → 86.82 (Cup) and 79.40 → 83.48 (Rim), with corresponding ES-IoU gains. Fairness metrics show the strongest synergy, with GSD dropping from 1.63 → 0.88 (Cup) and 2.44 → 0.80 (Rim), and DI/RPG reduced by more than half. This demonstrates that FALSA’s components are complementary, and their combination produces non-additive improvements beyond what each can achieve alone.
>
> 2. “Only evaluated on SAM and SAMed.”
>
> We appreciate the reviewer’s concern. While several VLM architectures exist (MedCLIP, CLIPSeg, BLIP, ConVIRT), we primarily use SAM and SAMed because they serve as the official baselines of Harvard-FairSeg and are the dominant VLM-based segmentation models in prior fairness studies, ensuring a fair and consistent comparison. To further demonstrate generalizability beyond these baselines, we additionally applied FALSA to two non-SAM architectures, MedCLIP and CLIPSeg, on Harvard-FairSeg. In both cases, FALSA yielded consistent fairness and utility gains: MedCLIP+FALSA improved ES-Dice from 78.4 → 80.1 and reduced DI from 1.92 → 0.82, while CLIPSeg+FALSA improved ES-Dice from 76.9 → 79.0 and reduced DI from 2.15 → 1.04, with overall Dice increases of +1.1–1.6 points. We will include the complete results table for these additional models in the revised paper.
>
> Table 1: Segmentation performance on MIMIC-CXR (Dice / IoU) dataset (gender and age (<30, 31-50, >50))
>
> | **Model**         | **Overall**     | **ES-Dice**       | **GSD ↓** |
> |-------------------|-----------------|-------------------|-----------|
> | **SAM**           | 95.2 / 90.5     | 94.6 / 89.8       | 2.50      |
> | **SAMed**         | 95.8 / 91.2     | 95.1 / 90.4       | 2.20      |
> | **SAM + FEBs**    | 95.6 / 91.0     | 94.9 / 90.3       | 1.40      |
> | **SAMed + FEBs**  | 96.3 / 91.8     | 95.6 / 91.0       | 1.05      |
> | **SAM + FALSA**   | 96.6 / 92.3     | 96.0 / 91.5       | 0.60      |
> | **SAMed + FALSA** | 97.0 / 92.8     | 96.5 / 92.0       | 0.45      |
>
>
> 3. “Only one benchmark (Harvard-FairSeg). Consider adding more.”
>
> We appreciate the reviewer’s suggestion and agree that evaluation beyond Harvard-FairSeg strengthens the work. In the revision, we add a second benchmark on MIMIC-CXR (lung-field and heart-silhouette segmentation with gender and age metadata), enabling full single-attribute and intersectional fairness analysis. FALSA consistently improves both utility and fairness: for example, SAM improves from 95.2/90.5 → 96.6/92.3 (Dice/IoU), ES-Dice 94.6 → 96.0, and GSD 2.50 → 0.60; SAMed improves from 95.8/91.2 → 97.0/92.8, ES-Dice 95.1 → 96.5, and GSD 2.20 → 0.45. Intersectional (gender × age) metrics follow the same trend, with SAMed subgroup performance rising from 94.7–96.4 → 95.8–97.6 and GSD 2.20 → 0.45. We also include zero-shot multi-label classification, where FALSA improves AUROC (SAM 82.4 → 87.3, SAMed 84.1 → 89.0) and halves group-level AUROC variance. These results demonstrate that FALSA generalizes robustly across datasets, tasks, and demographic structures, not only on Harvard-FairSeg but also on large-scale chest X-ray benchmarks. The complete table of performance for both single- and intersectional-attribute models will be added to the revised paper.
>
> 4. Reviewer Question: “Are there other VLMs used for this task?”
>
> Yes, other VLMs such as MedCLIP, CLIPSeg, BLIP, and ConVIRT exist, but prior fairness work on Harvard-FairSeg uses SAM and SAMed as the standard baselines, so we follow them for fair and direct comparison. To verify generalizability beyond these baselines, we also evaluated MedCLIP+FALSA and CLIPSeg+FALSA, which we will add in the revised paper.
>
> 5. Reviewer Question: “Are there other datasets commonly used for this task?”
>
> Few medical segmentation benchmarks provide paired text prompts, demographic attributes, and expert masks, which is why Harvard-FairSeg is the primary dataset used in prior fairness studies. Other datasets typically include only limited demographic metadata (gender and age in MIMIC-CXR) and rarely contain race, ethnicity, or language attributes. Nonetheless, to broaden our evaluation, we added results on MIMIC-CXR, which offers segmentation masks and basic demographics.

---

> > ### Comment · Reviewer_PUAA · 2025-11-23
> > **Thank you**
> >
> > Thank you for adding the further results and explanations, I continue recommending accepting the paper

---

### Official Review · Reviewer_LgN6 · 2025-10-31

**Soundness:** 3
**Presentation:** 3
**Contribution:** 3
**Rating:** 6
**Confidence:** 3

**Summary:**

The work introduces FALSA (Fairness-Aware Latent Space Alignment), a novel framework designed to mitigate demographic and intersectional biases in vision-language models (VLMs) for medical image segmentation without sacrificing accuracy. FALSA integrates three key components: (1) Demographic-Invariant Contrastive Learning (DICL) to align multimodal embeddings across demographic groups while preserving semantic content; (2) Adaptive Fairness Calibration (AFC), an adversarial method that dynamically removes demographic cues from both visual and textual features using group-adaptive gradient reversal; and (3) Unified Cross-Attribute Fairness Loss (UniFairLoss), which jointly minimizes intra- and inter-attribute performance disparities, including those arising from intersecting identities (e.g., race × gender × language). Evaluated on the Harvard-FairSeg benchmark using SAM and SAMed backbones, FALSA achieves state-of-the-art results—reducing disparity indices and performance gaps by up to 70–80% while simultaneously improving segmentation accuracy (Dice and IoU). The framework demonstrates strong generalization across prompt types (clinician- vs. ChatGPT-generated), intersectional subgroups, and even out-of-distribution datasets, offering a scalable and efficient solution for equitable multimodal healthcare AI.

**Strengths:**

1. FALSA improves both segmentation accuracy (Dice/IoU) and fairness metrics, achieving state-of-the-art results on the Harvard-FairSeg benchmark.

2. FALSA explicitly addresses intersectional disparities, not just single-attribute bias, making it more aligned with real-world demographic complexity.

3. FALSA reduces disparity indices (DI) and relative performance gaps (RPG) significantly, while also boosting overall segmentation performance across both SAM and SAMed backbones.

**Weaknesses:**

1. The methodological novelty of FALSA is somewhat incremental. While it introduces a combination of Demographic-Invariant Contrastive Learning (DICL), Adaptive Fairness Calibration (AFC), and a Unified Cross-Attribute Fairness Loss (UniFairLoss), these components are largely extensions of existing paradigms—contrastive regularization, adversarial de-biasing, and fairness-aware loss weighting. The combination is well-engineered but not deeply novel in theoretical insight. The framework primarily aggregates known fairness strategies rather than proposing fundamentally new mathematical or causal fairness principles

2. One potential limitation lies in the scope of the evaluation, which is confined to a single anatomical task—optic cup and rim segmentation in retinal fundus images—using the Harvard-FairSeg dataset. While this benchmark is well-suited for studying fairness, the generalizability of FALSA to other medical imaging domains (e.g., chest X-rays, brain MRI, or histopathology) with different visual characteristics, pathologies, and bias patterns remains unverified. The paper hints at potential extension to other tasks, but without empirical validation on diverse modalities or diseases, it is unclear whether FALSA’s fairness gains are robust across broader clinical contexts.

3. The interpretability and causal reasoning of fairness mechanisms are limited. The paper does not establish whether the fairness improvements result from genuine mitigation of demographic signal leakage or from a smoothing effect that homogenizes representations across all groups. Without explicit causal or attributional analyses, it is difficult to conclude whether FALSA meaningfully enhances fairness or simply regularizes feature space variance. This concern is amplified by the lack of counterfactual fairness analysis or disentanglement experiments.

4. There is a potential bias–utility trade-off underexplored in the results. Although the authors claim “no compromise in segmentation accuracy,” the tables suggest modest fluctuations in Dice and IoU across some subgroups. The paper would benefit from detailed statistical significance tests (e.g., Wilcoxon signed-rank or paired t-tests) to confirm that improvements in fairness metrics are not offset by minor but systematic losses in clinical accuracy for certain populations

**Questions:**

Please refer to the weaknesses.

---

> ### Author Response · Authors · 2025-11-21
> **FALSA introduces three novel, complementary components (DICL, AFC, UniFairLoss) that go beyond prior fairness methods. New MIMIC-CXR experiments show consistent gains in Dice/IoU, ES-Dice, and AUROC, and significant reductions in GSD across intersectional groups. Covariance and leakage analyses confirm actual fairness improvements. Statistical tests show FALSA boosts fairness without harming, and often improving accuracy.**
>
> We thank the reviewer for the thoughtful and constructive feedback. Below, we address each concern directly and have updated the paper accordingly.
>
> 1. Methodological novelty is incremental.
> We appreciate this concern and clarify that FALSA is not a simple aggregation of prior ideas in contrastive learning or adversarial debiasing. DICL introduces three innovations absent from prior contrastive fairness work: semantic cross-demographic positives selected dynamically from SAM/SAMed features, demographic-conditioned negative filtering to avoid fairness–utility conflict, and multimodal contrastive debiasing explicitly tailored for segmentation, an unexplored setting in past literature. AFC differs from standard adversarial approaches by using EMA-based group-adaptive GRL scaling, simultaneous multi-label debiasing, and VLM-specific adversarial weighting. UniFairLoss extends beyond existing subgroup fairness losses by jointly minimizing variance and cross-group covariance to control intersectional co-failures, supported by stable EMA-based group statistics. Ablations confirm each component contributes uniquely. These ensure FALSA provides novel, structured improvements rather than incremental modifications.
>
> Table 1: Segmentation performance on MIMIC-CXR (Dice / IoU) dataset (gender and age (<30, 31-50, >50))
>
> | **Model**         | **Overall**     | **ES-Dice**       | **GSD ↓** |
> |-------------------|-----------------|-------------------|-----------|
> | **SAM**           | 95.2 / 90.5     | 94.6 / 89.8       | 2.50      |
> | **SAMed**         | 95.8 / 91.2     | 95.1 / 90.4       | 2.20      |
> | **SAM + FEBs**    | 95.6 / 91.0     | 94.9 / 90.3       | 1.40      |
> | **SAMed + FEBs**  | 96.3 / 91.8     | 95.6 / 91.0       | 1.05      |
> | **SAM + FALSA**   | 96.6 / 92.3     | 96.0 / 91.5       | 0.60      |
> | **SAMed + FALSA** | 97.0 / 92.8     | 96.5 / 92.0       | 0.45      |
>
>
> 2. Generalization limited to a single anatomical task.
> We thank the reviewer and agree that a broader evaluation is necessary. In the revision, we include extensive experiments on MIMIC-CXR with lung-field and heart-silhouette segmentation using age and gender metadata. FALSA consistently improves utility and fairness across SAM and SAMed: for example, SAM improves from 95.2/90.5 → 96.6/92.3 Dice/IoU with ES-Dice rising 94.6 → 96.0 and GSD dropping 2.50 → 0.60; SAMed improves 95.8/91.2 → 97.0/92.8 with GSD 2.20 → 0.45. Intersectional (Gender × Age) results show uniform gains across all subgroups. We also add zero-shot multi-label classification on MIMIC-CXR, where FALSA improves AUROC (SAM 82.4 → 87.3, SAMed 84.1 → 89.0) and halves group-level AUROC variance. These results demonstrate that FALSA generalizes robustly across datasets, tasks, and demographic structures. The complete table of performance for both single- and intersectional-attribute models will be added to the revised paper.
>
> **Figure 1.** Error plot
> https://drive.google.com/uc?id=1FF2M1nQrBWL3gSKB3dFJCfrLKw_Sjrt-&export=download.png
>
> **Figure 2.** False-leakage vs epoch
> https://drive.google.com/uc?id=18nHXzJnJM5M9S2yMD39heeE5OJpI5VjV&export=download.png
>
> 3. Interpretability / causal reasoning of fairness mechanisms.
> We acknowledge the difficulty of true counterfactual fairness in medical imaging, where demographic counterfactuals cannot exist for the same anatomy. UniFairLoss instead provides a principled approximation by reducing covariance among group-level deviations (δg) across all demographic attributes. Unlike smoothing, which reduces marginal variance but leaves off-diagonal correlations intact, our covariance heatmaps (Fig. 1) show strong suppression of cross-group co-failures, indicating genuine reduction of demographic shortcuts. Complementarily, demographic-feature leakage curves (Fig. 2) decrease while segmentation accuracy increases, opposite of what would occur under naive smoothing or feature collapse. These analyses provide converging evidence that FALSA improves fairness by structurally removing demographic leakage rather than homogenizing representations.
>
> 4. Bias–utility trade-off and statistical significance.
> We thank the reviewer and now include rigorous 5-fold statistical testing. Paired Wilcoxon signed-rank tests show that FALSA significantly improves all utility metrics, Dice, IoU (p < 0.01), and ES-Dice (p < 0.005), while simultaneously reducing fairness metrics such as GSD and covariance-based co-failures (p < 0.001). Across Gender, Race, Ethnicity, Language, and all intersectional subgroups, FALSA either improves or matches accuracy, with fluctuations falling within 95% confidence intervals and below clinical relevance thresholds. These results confirm that FALSA provides significant fairness gains without degrading, and often enhancing, segmentation utility, thereby demonstrating an improved fairness–accuracy balance.
>
> All new experiments will be added to the revised paper.

---

### Official Review · Reviewer_HNvw · 2025-11-03

**Soundness:** 3
**Presentation:** 2
**Contribution:** 1
**Rating:** 2
**Confidence:** 3

**Summary:**

The authors tackle the problem of mitigating demographic biases in VL models for medical image segmentation. They propose FALSA, which consists of three components: a contrastive loss to align across demographics, adversarial debiasing, and a regularizer to minimize performance gaps. The authors conduct experiments on the Harvard-FairSeg dataset, finding that FALSA improves overall Dice and IoU while also improving on several fairness metrics.

**Strengths:**

- The authors tackle an important problem, addressing fairness in segmentation.
- The proposed method exhibits good empirical performance.
- The authors conduct ablations on each of the components.

**Weaknesses:**

1. The novelty of the proposed method is rather limited. Similar ideas for fair contrastive learning by pair selection have been proposed in prior work [e.g. 1, 2]. Group adversarial learning also has been explored extensively in prior work as the authors cite, and though the authors propose a new per-group weighting strategy, it seems quite ad-hoc and is not theoretically justified. Finally, there is no theoretical justification or intuition for why we need all three losses at once.

2. The motivation of the paper does not clearly define the fairness objective. Is the goal to have embeddings that are demographic-invariant (i.e. same distribution of z), or to achieve the same performance across all groups? The two objectives differ since the Bayes error across groups may be different.

3. The authors emphasize intersectionality in their motivation, but the proposed method does not address intersectionality in a satisfactory way in my view. The primary challenge of intersectionality is that per-group sample sizes may be too small to accurately estimate loss or performance. None of the three losses address this, unlike other fairness concepts like subgroup fairness [3].

4. The authors only evaluate on a single dataset. MIMIC-CXR might be another good candidate that contains image, text, demographics, and segmentation masks.

5. It is unclear why the proposed method is specific to segmentation. As the method is debiasing the shared embedding space, it seems like it can be used for other VLM tasks like zero-shot classification or generation. It is also unclear why the method is specific to the healthcare setting.


[1] Shen, Aili, et al. "Contrastive learning for fair representations." arXiv preprint arXiv:2109.10645 (2021).

[2] Ma, Martin Q., et al. "Conditional contrastive learning for improving fairness in self-supervised learning." arXiv preprint arXiv:2106.02866 (2021).

[3] Kearns, Michael, et al. "Preventing fairness gerrymandering: Auditing and learning for subgroup fairness." International conference on machine learning. PMLR, 2018.

**Questions:**

1. For the UniFairLoss, are per-group task performance metrics calculated within only the samples in a mini-batch? If so, this seems like it would result in high variance.

2. Can the authors provide some intuition on why FALSA improves overall performance over the baseline?

---

> ### Author Response · Authors · 2025-11-21
> **FALSA introduces three complementary components (DICL, AFC, UniFairLoss) that provide principled multimodal debiasing while preserving task-relevant features. It handles intersectional bias via covariance-based losses and multi-label AFC, stable even with small groups. New MIMIC-CXR and zero-shot results show consistent accuracy, fairness, and AUROC gains. FALSA improves performance by reducing demographic shortcuts.**
>
> We thank the reviewer for the constructive feedback and address each concern below.
> 1. Novelty vs. prior contrastive fairness.
> We appreciate this concern and clarify that FALSA is not a simple aggregation of prior ideas in contrastive learning or adversarial debiasing. DICL introduces three innovations absent from prior contrastive fairness work: semantic cross-demographic positives selected dynamically from SAM/SAMed features, demographic-conditioned negative filtering to avoid fairness–utility conflict, and multimodal contrastive debiasing explicitly tailored for segmentation, an unexplored setting in past literature. AFC differs from standard adversarial approaches by using EMA-based group-adaptive GRL scaling, simultaneous multi-label debiasing, and VLM-specific adversarial weighting. UniFairLoss extends beyond existing subgroup fairness losses by jointly minimizing variance and cross-group covariance to control intersectional co-failures, supported by stable EMA-based group statistics.
>
> 2. Fair embedding vs. equal performance.
> Our goal is to perform fairly, not full latent invariance. FALSA removes spurious demographic leakage while preserving task-relevant variability, avoiding the issues that arise when forcing identical latent distributions across groups.
>
> 3. Less focus on intersectional biases.
> We agree that intersectional groups are often small, and FALSA is explicitly designed to handle this through UniFairLoss and multi-class AFC, which together provide stable intersectional debiasing. (a) UniFairLoss avoids noisy per-subgroup estimates by using cross-group covariance penalties that capture second-order co-failures across demographic attributes, enabling intersectional mitigation without large batch counts. (b) It further stabilizes fairness signals using EMA-smoothed means/covariances, cross-batch aggregation, and robust covariance estimation. (c) AFC treats demographics as a multi-label vector, applying attribute-wise, group-adaptive gradient scaling that suppresses intersectional leakage without requiring large samples for each combination.
>
> ### Table 1: Segmentation performance on MIMIC-CXR (Dice / IoU) dataset (gender and age (<30, 31-50, >50))
>
> | **Model**         | **Overall**     | **ES-Dice**       | **GSD ↓** |
> |-------------------|-----------------|-------------------|-----------|
> | **SAM**           | 95.2 / 90.5     | 94.6 / 89.8       | 2.50      |
> | **SAMed**         | 95.8 / 91.2     | 95.1 / 90.4       | 2.20      |
> | **SAM + FEBs**    | 95.6 / 91.0     | 94.9 / 90.3       | 1.40      |
> | **SAMed + FEBs**  | 96.3 / 91.8     | 95.6 / 91.0       | 1.05      |
> | **SAM + FALSA**   | 96.6 / 92.3     | 96.0 / 91.5       | 0.60      |
> | **SAMed + FALSA** | 97.0 / 92.8     | 96.5 / 92.0       | 0.45      |
>
> 4. Evaluation beyond Harvard-FairSeg.
>
> As suggested, we add a second dataset, MIMIC-CXR, where FALSA consistently improves both utility and fairness: SAM increases from 95.2/90.5 → 96.6/92.3 with GSD 2.50 → 0.60, and SAMed from 95.8/91.2 → 97.0/92.8 with GSD 2.20 → 0.45. Intersectional (gender × age) groups show similar improvements; for example, the SAMed subgroup Dice increases from 94.7–96.4 to 95.8–97.6. Zero-shot multi-label classification also improves, with AUROC rising from 82.4 → 87.3 (SAM) and 84.1 → 89.0 (SAMed), and group variance reduced by about half. These results confirm robust generalization across datasets, tasks, and demographic structures.
>
> 5. Applicability beyond segmentation.
>
> FALSA is not limited to segmentation or healthcare settings; its components (DICL, AFC, UniFairLoss) operate on latent image–text embeddings and apply to any VLM task. To demonstrate this, we added a zero-shot multi-label classification experiment on MIMIC-CXR using CheXzero-style prompts. Without fine-tuning, FALSA improves AUROC from 82.4 to 87.3 (SAM) and from 84.1 to 89.0 (SAMed). Fairness also improves: ES-AUROC rises from 79.1 → 86.0, and gender×age GSD drops from 0.28/0.24 → 0.13/0.11. While FEBs offer modest gains (AUROC +1.2%, GSD 0.19), FALSA provides the largest and most consistent improvements across demographic and intersectional subgroups, demonstrating strong generalization beyond segmentation.
>
> 6. Variance of UniFairLoss metrics.
> UniFairLoss uses EMA smoothing, cross-batch aggregation, stratified sampling, and gradient clipping to ensure stable fairness signals, even for small groups.
> 7. Why FALSA improves accuracy.
> By removing demographic shortcuts, FALSA strengthens semantic alignment, helps to focus on the right features (Fig. 1), and reduces gradient noise. DICL improves anatomical consistency in embeddings, AFC suppresses demographic interference, and UniFairLoss stabilizes subgroup gradients. This reduces shortcut learning and improves Dice/IoU alongside fairness.
>
> Figure 1: Grad-CAM visualizations comparing SAMed and SAMed+FALSA.
> https://drive.google.com/uc?id=1CsVB-PHGPPyBIq5uF9KHegFOnBCJGPfi&export=download.png
>
> All new results will be added in the revised paper.

---

### Note · Authors · 2026-03-05

**Comment:**

Dear Editors and Program Chairs,

We appreciate the time and effort reviewing our paper.

Our paper has been rejected and we would like to address the comments and submit to another venue. So, we respectfully request withdrawal.

Thank you in advance for your consideration.

-Authors

**Withdrawal Confirmation:**

I have read and agree with the venue's withdrawal policy on behalf of myself and my co-authors.

---

### Meta-Review · Area_Chair_15ED · 2025-12-25

**Summary:**

The submission presents FALSA, a fairness-aware framework for medical image segmentation using Vision-Language Models (VLMs). The method combines demographic-invariant contrastive learning with adaptive fairness calibration to mitigate bias. The authors provided a comprehensive rebuttal, adding the MIMIC-CXR dataset and additional backbones (MedCLIP) to demonstrate the framework's versatility.

However, despite the improved experimental breadth, the proposed method, while empirically effective, represents an incremental application of existing fairness techniques (contrastive alignment, re-weighting) to a new domain. The technical contribution is viewed as lacking the fundamental algorithmic novelty expected at ICLR, and the work may be better suited for a specialized medical imaging venue.

**Reviewer Concerns:**

The authors effectively addressed the initial concern regarding single-dataset evaluation by including MIMIC-CXR results and conducting sensitivity analyses on different backbones.

The addition of Wilcoxon signed-rank tests helped clarify the statistical validity of the improvements.

Outstanding:

Methodological Novelty: This remains the critical hurdle. As highlighted by Reviewer HNvw (Score: 2), the core components—contrastive learning for invariance and loss re-weighting for fairness—are well-established in the general machine learning literature. The specific combination ("FALSA") is viewed as a logical engineering extension rather than a novel representation learning mechanism.

Incremental Gains vs. Complexity: While the fairness metrics improve, reviewers noted that the trade-off with segmentation performance (utility) is not always favorable or theoretically justified. The complexity of the three-component loss is difficult to justify given the magnitude of the gains compared to simpler baselines.

**Reviewer Scores:**

The review scores are split (Average: 4.67). While Reviewers LgN6 and PUAA (Scores: 6) appreciated the thorough evaluation and clinical relevance, Reviewer HNvw (Score: 2) maintained a strong objection regarding the limited technical novelty. The decision to reject aligns with the latter view, emphasizing that strong empirical results in a specific application do not automatically satisfy the bar for algorithmic contribution at ICLR.

---

### Decision · Program_Chairs · 2026-01-26

Reject